# Stoichiometry validation of supramolecular complexes with a hydrocarbon cage host by van 't Hoff analyses

**Toshiya M. Fukunaga** [1], **Yuzuka Onaka**[1], **Takahide Kato**[1], **Koki Ikemoto** [1] ✉ & **Hiroyuki Isobe** [1] ✉

Defining chemical processes with equations is the first important step in characterizing equilibria for the assembly of supramolecular complexes, and the stoichiometry of the assembled components must be defined to generate the equation. Recently, this subject has attracted renewed interest, and statistical and/or information-theoretic measures were introduced to examine the validities of the equilibrium models used during curve fitting analyses of titration. The present study shows that these measures may not always be appropriate for credibility examinations and that further reformation of the protocols used to determine the overall stoichiometry is necessary. Hydrocarbon cage hosts and their chloroform complexes formed via weak CH-π hydrogen bonds were studied, which allowed us to introduce van 't Hoff analyses for effective validation of the stoichiometries of supramolecular complexes. This study shows that the stoichiometries of supramolecular complexes should be carefully examined by adopting multiple measures with different origins.

Studies of chemical equilibria have provided important foundations of chemistry. The law of mass action allows us to characterize equilibria with equilibrium constants ($K$)[1–3], and decomposition analyses of Gibbs energy changes ($\Delta G$) allow us to quantitatively analyze chemical interaction energetics including $\Delta H$ and $\Delta S$[4,5]. Similarly, the chemistries of weak interactions embedded in elaborate molecular structures are being deepened and developed with studies of the equilibria of supramolecular complexes[6,7]. Although various interesting supramolecular systems are emerging, the weak interactions and complex structures often make it difficult to decide one of the most important characteristics of the equilibrium, i.e., the stoichiometry of the components involved in the equilibrium. Thus, traditionally, the stoichiometries of supramolecular complexes have been determined by the method of continuous variation using so-called Job plots of titration data, which allows us to identify a single equilibrium model for further analyses (Fig. 1a)[6,8–10]. However, since the credibility of Job plots was questioned in 2016[11,12], the development of alternative protocols to characterize equilibria has become an important subject particularly for supramolecular systems. An interesting proposal has recently been made for determining stoichiometries from fitting analyses (Fig. 1a). In this case, the titration data are fitted with multiple possible models with different stoichiometries (e.g. **1:1** vs. **1:2** in Fig. 1b), and the fitting data are then compared to assess credibility. A statistical method for the $F$-test using the $P$-value measure is useful to afford quantitative evaluations for the model comparison analyses, which has also been suggested by Hibbert/Thordarson among various measures[12–15]. More recently, we introduced an information-theoretic method using Akaike's information criterion (AIC) with the Akaike weight ($w_i$) measure as a more versatile method for comparison[16–19]. In the present investigation of CH-π hydrogen bonding, we noticed that the protocol needs further development and found that the van 't Hoff validation step should be used to examine the validity of the model comparison step. The stoichiometries of supramolecular complexes should be carefully examined by adopting multiple measures of different origins.

[1]Department of Chemistry, The University of Tokyo, Hongo 7-3-1, Bunkyo-ku, Tokyo 113-0033, Japan. ✉e-mail: kikemoto@chem.s.u-tokyo.ac.jp; isobe@chem.s.u-tokyo.ac.jp

**a**

**1928-2016**
1. Job plot
2. fitting analyses with <u>one</u> model
3. GOF evaluations
4. determination of $K_n$

**2016-2023**
1. fitting analyses with <u>multiple</u> models
2. GOF evaluations
3. model comparisons (F-test/P-value; AIC/$w_i$)
4. determination of $K_n$ of <u>one</u> model

**2023-**
1. fitting analyses with multiple models
2. GOF evaluations
3. model comparisons (F-test/P-value; AIC/$w_i$)
4. determination of $K_n$ of <u>multiple</u> models
5. van 't Hoff validations of <u>multiple</u> models

**b**

host + guest $\xrightarrow{K_1}$ **HG** **1:1** $\qquad \Delta\delta = \dfrac{\delta_{\Delta HG} K_1 [H][G]}{[G]_0}$
(H) (G)

$\Big\downarrow K_2$

**HG$_2$** **1:2** $\qquad \Delta\delta = \dfrac{\delta_{\Delta HG} K_1 [G] + 2\delta_{\Delta HG_2} K_1 K_2 [G]^2}{1 + K_1 [G] + K_1 K_2 [G]^2} \cdot \dfrac{[H]_0}{[G]_0}$

**Fig. 1 | Characterization of chemical equilibria. a** Protocols for characterization. **b** An example of the fitting equations used with NMR titration experiments. GOF goodness-of-fit.

## Results

### Synthesis of a hydrocarbon cage host

A problem was found in stoichiometry determinations during our studies of a cage-shaped hydrocarbon molecule. The hydrocarbon cage phenine polluxene (**1**) was recently synthesized as a phenine version of a minimal diamond twin cage[20-23], and host-guest complexation was noticed in this study of its structural diversification. The synthesis of an additional congener (**1a**, R = H; Fig. 2a) is described first. Following the preceding synthetic route for the $D_3$-symmetric phenine polluxene (**1b**, R = t-Bu), we started the synthesis from a decagonal macrocycle (**2**) composed of ten phenine units. After the introduction of two biphenyl arms via Suzuki-Miyaura coupling with **3**, the precursor was subjected to Ni-mediated Yamamoto coupling, which afforded the target polluxene (**1a**) in 49% yield. As shown in Fig. 2b, we determined the crystal structure of **1a**. The crystal structure showed a cage-shaped structure for the molecule, which revealed the presence of multiple chloroform molecules trapped inside the cage[24-26]. Some chloroform molecules were found at nearly identical positions in the case of the $D_3$-symmetric congener (**1b**)[20] (Supplementary Fig. 2). For instance, one chloroform molecule was found at the center of the cage, which indicated the presence of CH·π hydrogen bonding between them. Based on our interest in weak hydrogen bonds with CH·π contacts[27-33], we then investigated solution-phase assembly of the phenine polluxene and chloroform complex.

### Titration experiments and fitting analyses

Titration experiments were performed to study the association equilibrium for polluxene and chloroform. Thus, to a solution of **1a** (0.503 mM) in deuterated cyclohexane (C$_6$D$_{12}$) was gradually added a solution of chloroform in C$_6$D$_{12}$ (1.01 M) at 298 K, and the ¹H NMR spectra were recorded. The resonance for chloroform showed upfield shifts upon mixing with **1a**, and the changes ($\Delta\delta$) were plotted against the host-guest ratios CHCl$_{30}$/**1a**$_0$ (Fig. 3a). The titration was performed in triplicate, and fitting analyses were performed with all of the data to obtain association constants ($K_n$) by using equations for **1:1** and **1:2** (see Fig. 1b and Supplementary Information for detailed procedures). Both equilibrium equations resulted in acceptable goodness-of-fit (GOF)

**a**

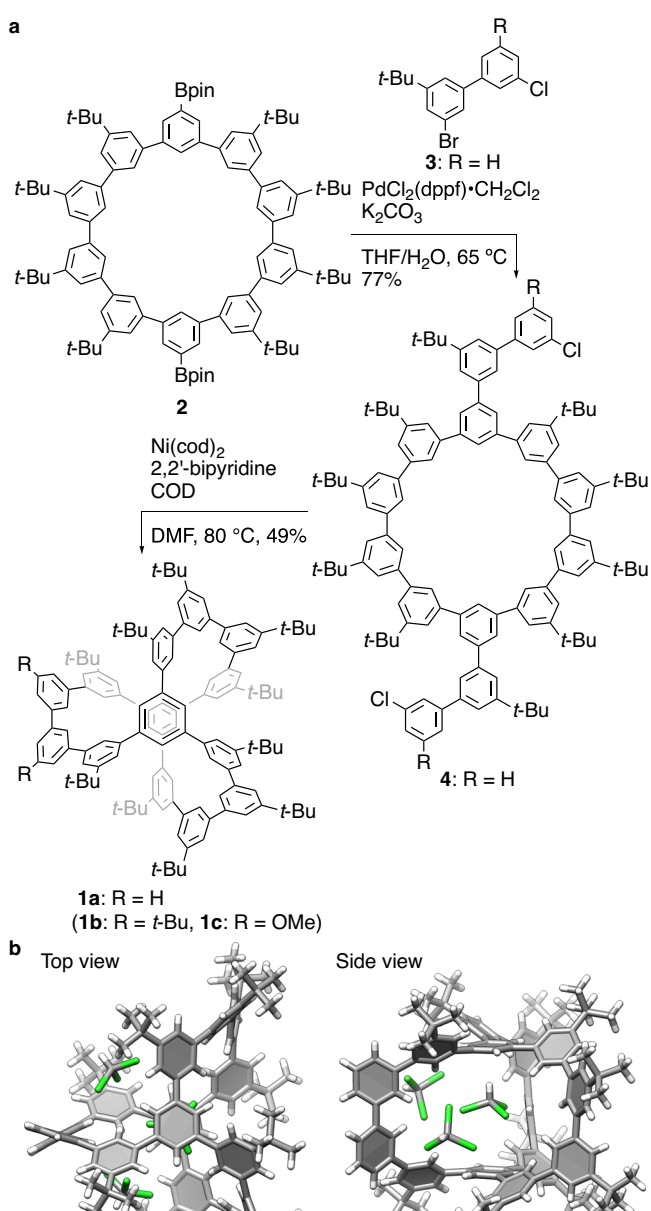

**b** Top view Side view

**Fig. 2 | Phenine polluxene 1. a** Synthesis of **1a**. **b** Crystal structure of **1a**. Three chloroform molecules were identified, and highly disordered residues were removed by the SQUEEZE protocols (Supplementary Fig. 2). The chloroform molecule at the center had two up and down orientations, and the up-oriented molecule is shown for a representative molecule with 50% occupancy. See Supplementary Fig. 2 for a detailed comparison with **1b**. dppf 1,1'-bis(diphenylphosphino)ferrocene, THF tetrahydrofuran, Bu butyl, COD 1,5-cyclooctadiene, DMF N,N-dimethylformamide, Me methyl.

levels with determination coefficients of $R^2 > 0.99$ for the fit. We first compared fitting credibility with F-tests using the P-values[12]. The F-test compares the credibilities of two models (**1:1** vs. **1:2**), and a small P-value below 0.05 indicates that the more complex model (**1:2**) is more credible[13]. As shown in Fig. 3a for **1a** as the host, a P-value on the order of $10^{-5}$ was obtained. We thus concluded that for the equilibrium between **1a** and chloroform, **1:2** was more credible than **1:1**. The fitting credibility was also examined with the information-theoretic measure of AIC[16], and high $w_i$ values were obtained for **1:2** (0.9165 with **1a** and 0.9405 with **1b**) with minute supports for **1:1** ($w_i < 0.1$)[17]. Thus, two

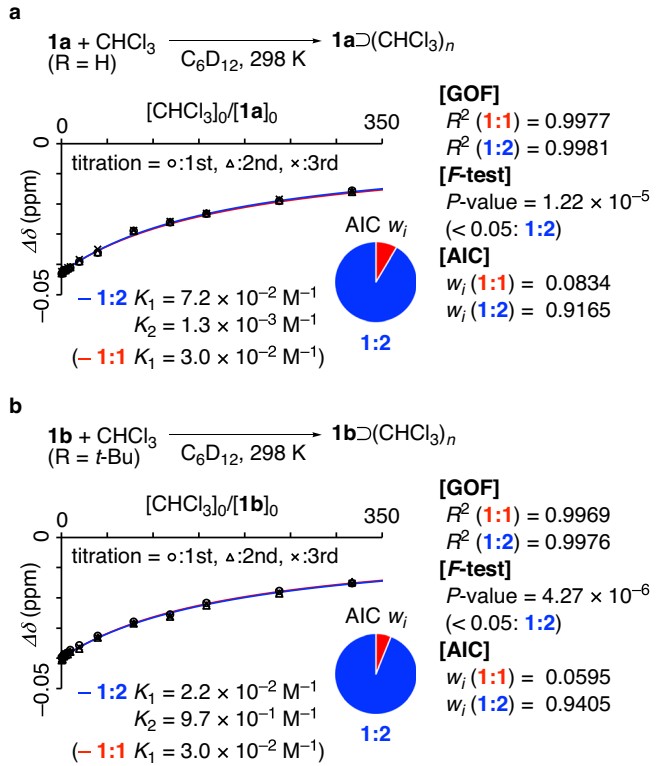

**Fig. 3 | Titration experiments for solution-phase assembly of $1 \supset (CHCl_3)_n$ complexes.** The triplicate titration data were fitted with two equilibrium models for **1:1** (red curve) and **1:2** (blue curve). The fit quality was evaluated with $R^2$-values indicating the GOF, and the model credibility was examined with two measures: $P$-values for the $F$-test and $w_i$ values for the AIC. The blue and red lines show fitting curves for **1:2** and **1:1**, respectively. **a** Experiments with **1a** and chloroform. **b** Experiments with **1b** and chloroform. GOF goodness-of-fit, AIC Akaike's information criterion, Bu butyl.

different fitting analyses supported the credibility of **1:2** for the $1a \supset (CHCl_3)_n$ complex. Similarly, the association equilibrium between **1b** and chloroform was examined by titration. As summarized in Fig. 3b, the credibility of **1:2** for the $1b \supset (CHCl_3)_n$ complex was also supported by both the $F$-test and the AIC.

### van 't Hoff validation of model credibility

During our examinations of the two-stage association constants for the $1 \supset (CHCl_3)_2$ complexes, however, we noticed unreasonable discrepancies, which prompted us to investigate model validation further. Thus, for two phenine polluxene complexes with different substituents (**1a**: R = H and **1b**: R = *t*-Bu), the first-stage association constants ($K_1$) were comparable at $10^{-2}$ M$^{-1}$, but the second-stage association constants ($K_2$) were significantly different for **1a** and **1b** and showing a difference of nearly two orders of magnitude ($10^{-3}$ M$^{-1}$ vs. $10^{-1}$ M$^{-1}$; Fig. 3). Considering the subtle structural differences at the outer surfaces of the cages, we found this discrepancy chemically unreasonable and started further verification of the models selected for stoichiometry determination. After several trials with alternative measures, we found that the introduction of a thermodynamic verification measure provided an additional criterion for model selection. Thus, we performed triplicate titration experiments at 6 different temperatures (283, 288, 298, 308, 318, and 328 K; Supplementary Fig. 3), and the association constants from the fitting analyses were plotted in $\ln K_n$ vs. $1/T$ graphs for the van 't Hoff analysis[2,3,34,35]. For both **1a** and **1b** with the 1:2 stoichiometry, the data were fitted with the van 't Hoff equation. The fitting equations with statistical factors are $\ln K_1 = -\Delta H_1 \cdot (1/T) + \Delta S_1/R$ and $\ln K_2 = -\Delta H_2 \cdot (1/T) + \Delta S_2/R - \ln 2$ where $R$ is

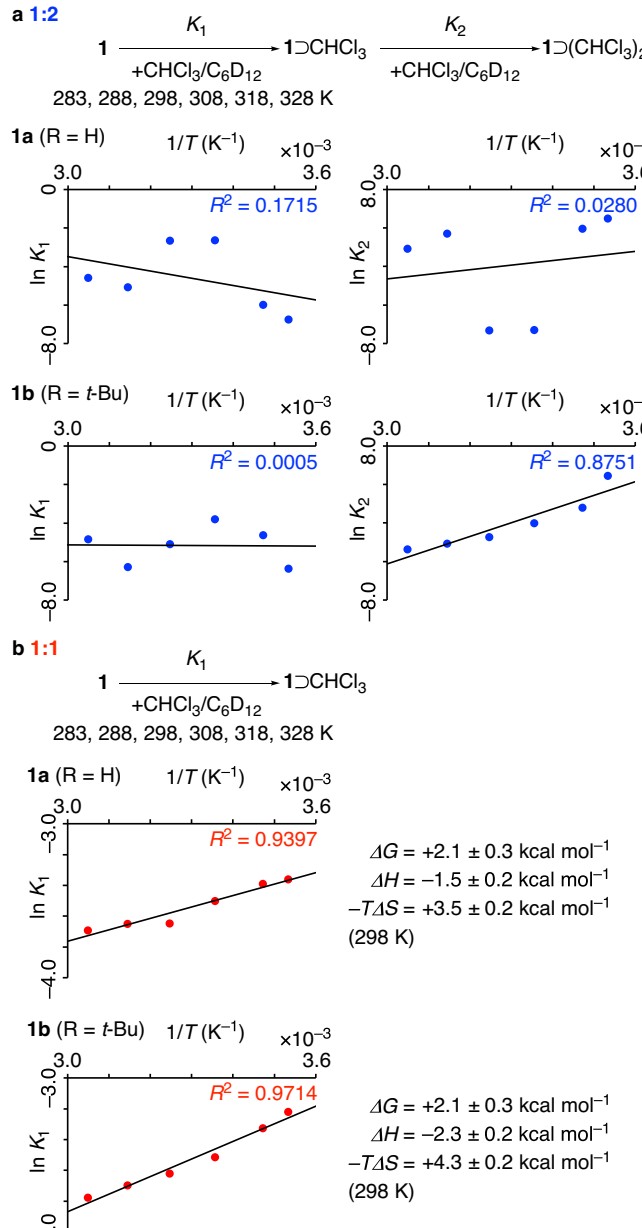

**Fig. 4 | van 't Hoff validation of variable-temperature titration data for the $1 \supset (CHCl_3)_n$ complexes. a** Validation of the 1:2 stoichiometry. **b** Validation of the 1:1 stoichiometry. Thermodynamic parameters from the van 't Hoff fits are shown.

the gas constant[14,36,37]. As shown in Fig. 4a, the $R^2$ values for the van 't Hoff fits were mostly poor, ranging from 0.0005 to 0.8751. These values showed that the thermodynamics determined with the van 't Hoff analyses disproved **1:2** as the stoichiometry model for the $1 \supset (CHCl_3)_n$ complex. We then performed the same van 't Hoff validation for **1:1**. As shown in Fig. 4b, we obtained linear fits with high $R^2$ values of 0.9397 and 0.9714 for **1a** and **1b**, respectively. With this result from the van 't Hoff validation, we therefore concluded that phenine polluxene and chloroform formed **1:1** complexes in solution. Further support for this conclusion was derived from theoretical analyses of the $1 \supset (CHCl_3)$ complex.

### Theoretical studies of the 1:1 complex

We performed a theoretical analysis of the $1 \supset (CHCl_3)$ complex with density functional theory (DFT) calculations for **1a** with Me substituents as models for the *t*-Bu substituents. As shown in Fig. 5, the

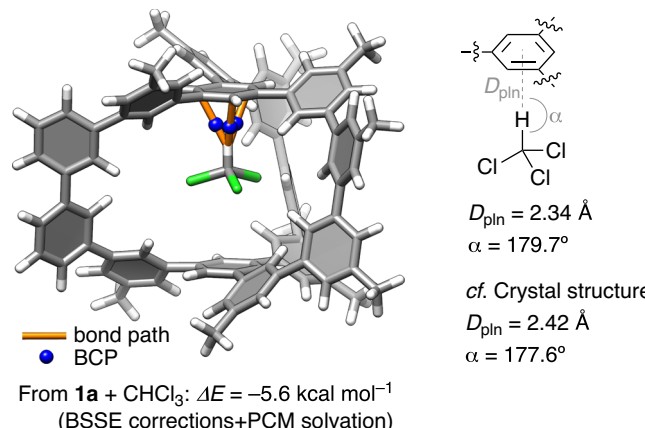

bond path
BCP

$D_{pln}$ = 2.34 Å
$\alpha$ = 179.7°

*cf.* Crystal structure
$D_{pln}$ = 2.42 Å
$\alpha$ = 177.6°

From **1a** + CHCl₃: $\Delta E$ = −5.6 kcal mol⁻¹
(BSSE corrections+PCM solvation)

**Fig. 5 | A DFT structure of the 1⊃(CHCl₃) complex.** The structure was optimized with DFT calculations at the LC-BLYP/6-311 G(d) level of theory with phenine polluxene containing methyl substituents as a model for **1a**. The association energy was estimated after counterpoise BSSE corrections in the presence of PCM for cyclohexane. Quantum mechanical AIM analyses revealed the presence of bond paths and (3,−1) BCP. The CH-π distances ($D_{pln}$) and CH-π angles ($\alpha$) are shown for the theoretical model and the crystal structure. BCP: bond critical point, BSSE: basis-set superposition error. PCM: polarizable continuum model.

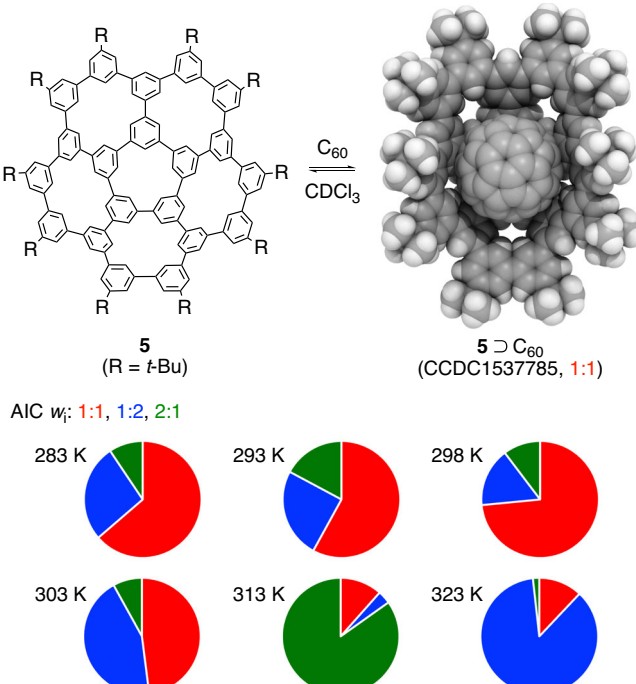

**5**
(R = *t*-Bu)

**5 ⊃ C₆₀**
(CCDC1537785, 1:1)

AIC $w_i$: 1:1, 1:2, 2:1

283 K 293 K 298 K

303 K 313 K 323 K

**Fig. 6 | An assessment study with a ball-in-bowl nanocarbon assembly (5⊃C₆₀): AIC.** The $w_i$ values for three models (1:1, 1:2 and 2:1) showed that all the models could be supported. See Supplementary Fig. 11 for further details of fitting analyses. Bu butyl.

structure of **1:1** was obtained from a geometry optimization exhibiting good convergence, whereas the structure of **1:2** with two chloroform molecules did not converge. The association energy of the **1⊃(CHCl₃)** complex was estimated as $\Delta E$ = −5.6 kcal mol⁻¹ at the LC-BLYP/6-311 G(d) level of theory[38,39] via counterpoise basis-set superposition error (BSSE) corrections[40] in the presence of the polarizable continuum model (PCM)[41] for cyclohexane, which reasonably reproduced the experimental association enthalpy of $\Delta H$ = −1.5 kcal mol⁻¹[42,43]. As detailed in Supplementary Information, large $|\delta_{\Delta HG}|$ values from fitting analyses indicated the presence of unknown origins in experimental values, which might be one of the reasons of the minor difference. The DFT electron density was also subjected to quantum mechanical atoms-in-molecule (AIM) analyses[44], which revealed three intermolecular bonding paths for the host and guest (Fig. 5). The bond paths possessed (3,−1) bond critical points (BCPs), which indicated the presence of CH-π hydrogen bonds between the hydrocarbon cage and chloroform[27,29,45–47]. The relative orientations of the chloroform and a phenine panel in the calculated structure were analyzed to show a CH-π distance ($D_{pln}$) of 2.34 Å and a CH-π angle ($\alpha$) of 179.7°[27,48]. These geometry parameters matched well with those of the crystal structure (see also Fig. 2b). Thus, the theoretical calculations confirmed that the 1:1 stoichiometry provided a reasonable model for the **1⊃(CHCl₃)ₙ** complex that was uniquely formed by CH-π hydrogen bonding between the hydrocarbon cage and the guest.

## Scope: Applicability of the van ′t Hoff validation

Finally, we examined the scope of the van ′t Hoff validation method by assessing its applicability to other systems. Among various host-guest combinations of nanocarbon molecules[23,49], we have previously studied a ball-in-bowl assembly between C₆₀ and phenine 5circulene (**5**) by performing variable-temperature titration experiments with NMR spectroscopy[50,51]. The triplicate titration data comprising 252 ¹H NMR spectra were thus subjected to re-analysis for the fitting analyses, and the fitted data for 1:1, 1:2 and 2:1 models were first compared by using the $w_i$ values from AIC analyses[16]. As shown in Fig. 6, although moderate preference was found for a chemically reasonable 1:1 model that was observed in the crystal structure, the $w_i$ values did not consistently support one model over the 6 variable-temperature conditions.

We then subjected the fitted data to the van ′t Hoff validations by plotting the K values in the 1/T-ln K graphs. As shown in Fig. 7, the 1:1

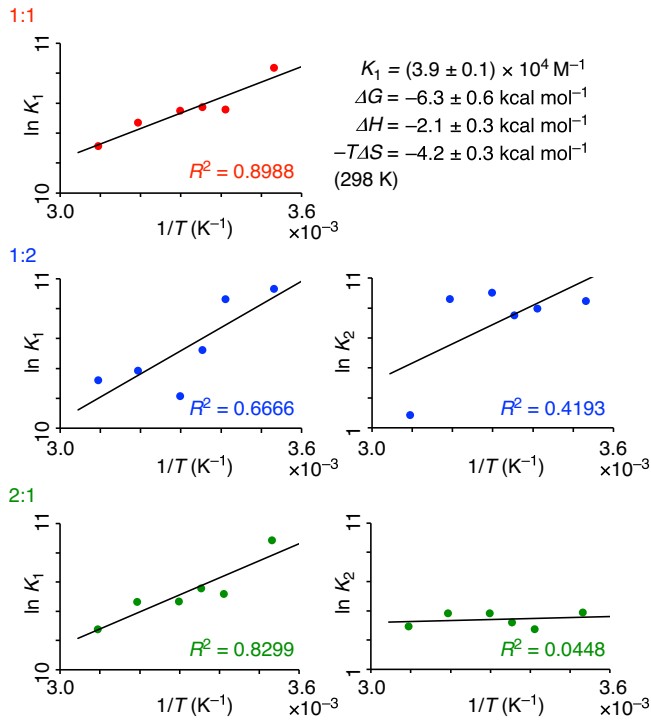

1:1

$K_1$ = (3.9 ± 0.1) × 10⁴ M⁻¹
$\Delta G$ = −6.3 ± 0.6 kcal mol⁻¹
$\Delta H$ = −2.1 ± 0.3 kcal mol⁻¹
$−T\Delta S$ = −4.2 ± 0.3 kcal mol⁻¹
(298 K)

$R^2$ = 0.8988

1:2

$R^2$ = 0.6666 $R^2$ = 0.4193

2:1

$R^2$ = 0.8299 $R^2$ = 0.0448

**Fig. 7 | An assessment study with a ball-in-bowl nanocarbon assembly (5⊃C₆₀): van ′t Hoff validation.** The $K_n$ values from the fitting analyses with three models (1:1, 1:2 and 2:1) were respectively plotted in the 1/T-ln K graph.

model showed a linear relationship with a high $R^2$ value of 0.8968, whereas other models (1:2 and 2:1) failed to show consistent linear relationships. Considering the observation of the 1:1 crystal structure of **5⊃C₆₀** (Fig. 6), we concluded that the van ′t Hoff validation method

was applicable to this ball-in-bowl system that was tightly assembled with a $K$ value of $(3.9 \pm 0.1) \times 10^4 \, M^{-1}$ (298 K). The results showed that the van 't Hoff validation method can be versatile to be applied to many other supramolecular systems.

## Discussion

In summary, we investigated measures and methods for determining the stoichiometries of supramolecular complexes. The results obtained from careful analyses of 954 $^1$H NMR spectra showed that drawing conclusions about the stoichiometry from a single measure is risky, and we propose the following procedure for stoichiometry analyses. 1. Fit the titration data with multiple models. 2. The GOF for the fit should be evaluated with the $R^2$ value. 3. The credibilities of model stoichiometries should be examined with quantitative measures such as $P$-values and/or $w_i$ values. 4. The $K_a$ values derived from the step 1 should be examined ideally by comparing congeners and structural models. 5. Model comparisons should be validated by performing van 't Hoff analyses. When the van 't Hoff validations indicate non-linear relationships in the $1/T$-ln $K$ graph, one may also need to check the existence of temperature-dependent enthalpy, which should give rise to new subjects to be investigated[2,3,34,52]. 6. By examining the results from the necessary measures and steps, and probably with the aid of theoretical calculations, a chemically sound stoichiometry should be concluded. In the present study, we found that a phenine polluxene hydrocarbon cage served as an interesting host for a small molecule such as chloroform via formation of CH-π hydrogen bonds. Further structural diversifications are currently under investigation, and this study showed that host-guest complexations should also be considered for the development of unique functions.

## Methods

### Syntheses

The syntheses of the precursors are described in Supplementary Information, and the final cyclization step for **1a** is described here. A mixture of 2,2'-bipyridine (848 mg, 5.43 mmol), 1,5-cyclooctadiene (668 μL, 5.43 mmol) and Ni(cod)$_2$ (1.49 g, 5.43 mmol) was stirred in DMF (11.3 mL) at 80 °C for 30 min. A solution of **4** (115 mg, 67.8 μmol) in DMF (22.6 mL) was then added dropwise over 120 min, and the mixture was stirred at 80 °C for 2 h. After the mixture was cooled down to ambient temperature, 2 M aq. HCl (30 mL) was added. The organic layers was separated, and the aqueous layer was extracted with chloroform (20 mL × 3). The combined organic layer was washed with saturated aq. NaHCO$_3$ (50 mL), dried over Na$_2$SO$_4$ and concentrated in vacuo. The crude material was purified by gel permeation chromatography (GPC) to afford **1a** in 49% yield (54.0 mg, 33.2 μmol). $^1$H NMR (600 MHz, CDCl$_3$) δ 7.96 (s, 2H), 7.93 (s, 2H), 7.91 (s, 4H), 7.79 (s, 2H), 7.78 (s, 4H), 7.76-7.74 (m, 10H), 7.65-7.63 (m, 6H), 7.60-7.58 (m, 6H), 7.57 (s, 4H), 7.56 (s, 4H), 1.49 (s, 36H), 1.37 (s, 36H), 1.36 (s, 18H); $^{13}$C NMR (151 MHz, CDCl$_3$) δ 152.4, 152.3, 152.2, 143.3, 143.1, 143.1, 142.9, 142.8, 142.3, 141.3, 141.1, 141.1, 140.9, 129.4 (CH), 127.7 (CH), 126.2 (CH), 125.8 (CH), 125.8 (CH), 125.2 (CH), 124.8 (CH), 124.6 (CH), 124.5 (CH), 124.3 (CH), 124.0 (CH), 123.8 (CH), 123.7 (CH), 123.4 (CH), 123.1 (CH), 35.2, 35.1, 35.1, 31.6 (2 overlapping CH$_3$), 31.6 (CH$_3$); HRMS (MALDI-TOF) ($m/z$): M$^+$ calcd. for C$_{124}$H$_{134}$ 1623.0480, found 1623.0497. Chromatograms and spectra are included in Supplementary Information.

### X-ray crystallographic analysis

A single crystal was grown from a solution of **1a** in chloroform by slowly diffusing methanol vapor at 25 °C. The crystal was mounted on a thin polymer tip with cryoprotectant oil and frozen via flash cooling. The diffraction experiment was carried out at 95 K with a synchrotron X-ray source at the BL17A beamline equipped with a Dectris EIGER X 16 M PAD detector (KEK Photon Factory), and the diffraction data were processed with XDS[53]. The structure was solved by direct methods with SHELXT[54] and refined by full-matrix least-squares on $F^2$

using SHELXL-2018/3[55] running with Yadokari-XG 2009[56]. In the refinements, the $t$-Bu groups and solvent molecules were restrained by SIMU, DELU, ISOR, DFIX and DANG. The nonhydrogen atoms were analyzed anisotropically, and hydrogen atoms were input at the calculated positions and refined with a riding model. Some solvent molecules could not be modeled due to severe disorder, and the residual density was eliminated by using the PLATON/SQUEEZE protocol[57,58]. Crystal and structure refinement data are included in Supplementary Information.

### Titration experiments and fitting analyses

A solution of **1a** (5.63 mg, 3.47 μmol/Precision balance XPE205V, Mettler Toledo) was prepared in C$_6$D$_{12}$ (0.690 mL; 5.02 mM/Gastight syringe1001, Hamilton), and a solution of chloroform (242 mg, 2.02 mmol/XPE205V) was prepared in C$_6$D$_{12}$ (2.00 mL; 1.01 M/Gastight syringe1001). To the solution of **1a** in an NMR tube (0.500 mL/Gastight syringe1001) was added 1.01 M solution of chloroform (+1.00 μL/Gastight syringe1701), and $^1$H NMR spectra were respectively recorded at 283 K, 288 K, 298 K, 308 K, 318 K and 328 K. When the temperature was adjusted, the sample was maintained at the temperature for approximately 15 min before recording the spectrum. The spectra were subjected to fourfold zero filling to secure a digital resolution of 0.013 Hz (6.6 kHz range/512k points). The NMR titration experiments were performed by adding additional chloroform solution (+3.00 μL, +4.00 μL, +4.00 μL, +6.00 μL, +7.00 μL/Gastight syringe1701, +25.0 μL, +50.0 μL/Microsyringe MSGFN50, Ito, +100 μL, +100 μL, +100 μL, +200 μL and +200 μL/Gastight syringe1725, and these were repeated three times to generate triplicate titration data. The spectra were recorded on JEOL RESONANCE JNM-ECA II 600 spectrometer equipped with a temperature-controlled UltraCOOL probe (error: ±0.1 °C). A solution of **1b** (8.21 mg, 4.73 μmol) was also prepared in C$_6$D$_{12}$ (0.940 mL; 5.03 mM), which was subjected to identical titration experiments with a solution of chloroform (238 mg, 1.99 mmol) in C$_6$D$_{12}$ (2.00 mL; 1.00 M). As a third entry of the cage host, **1c** was subjected to the titration experiment by using **1c** (5.91 mg, 3.51 μmol) in C$_6$D$_{12}$ (7.00 mL; 5.01 mM) and chloroform (240 mg, 2.01 mmol) in C$_6$D$_{12}$ (2.00 mL; 1.01 M). The chemical shift of chloroform was measured by using a singlet peak from C$_6$H$_{12}$ at 1.38000 ppm as a reference, and the changes in the chemical shifts ($\Delta\delta$) were derived by using a chemical shift of unbound free chloroform (7.11590 ppm) as a standard. In total, 702 $^1$H NMR spectra were recorded and used for the analyses. All the source data for the fitting analyses were provided in Supplementary Information (excel files). The data were plotted against CHCl$_{30}$/**1**$_0$ (Fig. 3), where CHCl$_{30}$ and **1**$_0$ are the total concentrations of the guest and host. To derive the association constants ($K_n$), the titration data were fitted with the following two equations[6]:

$$\Delta\delta = \frac{\delta_{\Delta HG} K_1 [H][G]}{[G]_0} \qquad (1)$$

$$\Delta\delta = \frac{\delta_{\Delta HG} K_1 [G] + 2\delta_{\Delta HG_2} K_1 K_2 [G]^2}{1 + K_1 [G] + K_1 K_2 [G]^2} \cdot \frac{[H]_0}{[G]_0} \qquad (2)$$

where H is the host (**1**) and G is the guest (chloroform) in the **1:1** and **1:2** equilibrium models, respectively, and OriginPro 2023 (OriginLab Corporation) was used for fitting. The experimental $^1$H NMR spectra with resonances derived from the fitting analyses are shown in Supplementary Fig. 4 (**1:1**) and Supplementary Fig. 5 (**1:2**), which showed a higher consistency between **1a** and **1b** for the 1:1 model. The fitted data were analyzed with GOF/$R^2$ values, $F$-test/$P$-values and AIC/$w_i$ values, and the fitting data as well as $K_n$ values are included in Supplementary Information along with the equations and codes. To derive the determination coefficients, the temperature-dependent $K_n$

values were plotted in $1/T$-ln $K_n$ graphs and fitted with the van 't Hoff equations with statistical factors considered for one-site hosts: ln $K_1 = -\Delta H_1 \cdot (1/T) + \Delta S_1/R$ and ln $K_2 = -\Delta H_2 \cdot (1/T) + \Delta S_2/R - \ln 2$[14,37]. To confirm the repeatability of the titration experiments, independent titration data at 298 K were separately fitted to derive averaged $K_n$ values with standard deviations (SD) of three titrations as follows ($K_n \pm$ SD). **1a**⊃(CHCl$_3$)$_1$: $K_1 = (3.3 \pm 0.2) \times 10^{-2}$ M$^{-1}$; **1a**⊃(CHCl$_3$)$_2$: $K_1 = (3.8 \pm 4.4) \times 10^{-2}$ M$^{-1}$, $K_2 = (8.4 \pm 6.1) \times 10^{-3}$ M$^{-1}$; **1b**⊃(CHCl$_3$)$_1$: $K_1 = (3.0 \pm 0.2) \times 10^{-2}$ M$^{-1}$; **1a**⊃(CHCl$_3$)$_2$: $K_1 = (1.1 \pm 0.4) \times 10^{-2}$ M$^{-1}$, $K_2 = 9.6 \pm 9.8$ M$^{-1}$. The SD values are reasonably low particularly for **1:1** models. We also analyzed the fitted chemical shifts for **1:1** and **1:2** both with **1a** and **1b** as the host (see Supplementary Information). The results with **1c** were summarized in Supplementary Figs. 6–9.

### Theoretical studies with DFT calculations

DFT calculations at the LC-BLYP/6-311 G(d) level[38] of theory were performed with Gaussian 16[59]. For the polluxene hosts, the $t$-Bu substituents were replaced by Me groups in the model. Geometry optimizations were performed for **1**, chloroform and the **1**⊃(CHCl$_3$) complex in the presence of cyclohexane PCM[41], and the association energies were estimated with BSSE corrections[40]. Quantum mechanical AIM analyses were performed with Multiwfn[60]. The results are summarized in Fig. 5 and Supplementary Fig. 12, and the cartesian coordinates are provided as Source Data.

### Assessment study of the van 't Hoff validation method via re-analyses

The titration data from the previous report of **5**⊃C$_{60}$ were used for the re-analyses[51]. For the titration experiment, **5** and C$_{60}$ were mixed in CDCl$_3$ at 14 different ratios for triplicate titrations under 6 different temperature conditions to record $14 \times 3 \times 6 = 252$ spectra (Supplementary Fig. 10). A $^1$H resonance of **5** at the most downfield region of 9.008 ppm ($\delta_{\text{unbound}}$) was used as a reference, and the $\Delta\delta$ values were respectively fitted with three different models (1:1, 1:2 and 2:1 models) (Supplementary Fig. 11). At the stage 3, AIC analyses were performed to obtain $w_i$ values (Fig. 6), which failed to support one single model. The temperature-dependent $K$ values were then plotted in $1/T$-ln $K$ graphs for the van 't Hoff validations (Fig. 7). The van 't Hoff validation supported the 1:1 model by showing a linear relationship in the graph, which matched well with the structure observed in the crystal (see Fig. 6). In the previous study, an averaged aromatic resonance values were used for the fitted analyses, and minor differences were found for the $K$ values and thermodynamic parameterers[51].

## Data availability

Crystallographic data are available at Cambridge Crystallographic Database Centre (https://www.ccdc.cam.ac.uk) as CCDC2281309. Source data (titrations, fitted graphs and cartesian coordinates from DFT calculations) are provided at figshare (https://doi.org/10.6084/m9.figshare.24309472). All other data that support the findings of this study are available from the corresponding author.

## Code availability

Codes for fitting analyses on OriginPro are provided in Supplementary Information.

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

## Acknowledgements

We were granted access to X-ray diffraction instruments in KEK (BL17A, no. 2022G596). This work was partly supported by KAKENHI (20H05672, H.I.; 22H02059, K.I.; 22K20527, T.M.F.) and JST ACT-X (JPMJAX23DI, T.M.F.). Y.O. thanks MERIT-WINGS program for predoctoral fellowship.

## Author contributions

T.M.F. and T.K. performed experiments (synthesis and titrations); T.M.F., Y.O., K.I. and H.I. examined model credibility; T.M.F. performed theoretical calculations; T.M.F, Y.O., K.I. and H.I. prepared figures; T.M.F, Y.O., K.I. and H.I. discussed the content, and H.I. wrote the manuscript; T.M.F, Y.O., K.I. and H.I. contributed to the editing of the manuscript and the Supplementary Information.

## Competing interests

The authors declare no competing interests.
