## [Peer review file · Nature Communications]

REVIEWER COMMENTS

Reviewer #1 (Remarks to the Author):

In this manuscript, Isobe et al propose a new, rigorous workflow to examine the validity of equilibrium models. They put forward van't Hoff analyses for effective validation of stoichiometry when investigating the hydrocarbon cage hosts and their chloroform complexes. The authors have synthesized an intriguing molecule and described its encapsulation mechanism with chloroform beautifully. Apart from experimental observations, theoretical studies were carried out to further confirm the stoichiometry of these complexes. Overall, this work is intriguing, all experiments were well conducted and the conclusion drawn is an important one – we indeed need to be critical of our findings with only one type of measurement for stoichiometry experiments.

For works of methodology like the one here, there is naturally a question of novelty. However, it is my strong belief that proposing universal methods that are simple to grasp and execute has tremendous value to the scientific method. I believe the work to be thorough, the method well described, and recommend publication in principle. Nevertheless, I see a few flaws that I would like to get the author's comments on:

1. Universality. Their current manuscript demonstrates the method on «only two» host-guest complexes. In addition, the result from this manuscript has provided a counter-example to the previous paper published by the same group early this year (Angew. Chem. Int. Ed.2023,62, e2022190). I do think the manuscript would benefit from a few additional examples, even if it is reanalyses of previously published manuscripts – from the authors or others.
2. Methodology. I am not entirely convinced that I need to follow the workflow proposed in Fig.1. Why do we have to go for model comparisons (F-test/P-value; AIC/wi)? In the cases presented in this manuscript, model comparisons did not help and gave the wrong conclusion. I think it should be made abundantly clear whether F/P-tests of AIC/wi should always be conducted or abandoned in favor of van't Hoff validation. In case of contradicting conclusions, it should also be made clear, under which circumstances which approach should be given more consideration.

Reviewer #2 (Remarks to the Author):

This paper introduces a new approach to deal with one of the most difficult challenges in the field of host-guest binding studies – namely “model selection” or selecting which binding model describes best the data being analyzed in supramolecular titration studies.

This topic is of on-going interest and debate in the field and given that the accurate determination of binding constants and real stoichiometry in supramolecular underpins any attempts for the rational develop of better host-guest systems, this topic is clearly of interest to the readership of Nature Communication.

The authors key insight and contribution in this paper resides in using van't Hoff analysis for validation of binding stoichiometries.

I do think that this approach could be a very valuable tool in the field but there are a few issues in the current manuscript that first needed be addressed before the paper is published.

1. To start with, the historical overview in the introduction is very useful, including Figure 1. But there are few points that may need adjustment, particular in relation to the 2016-2023 period:

1a) The authors frequently cite the Thordarson/Hibbert paper from 2016 and the F-test discussed in that review. But they don't mentioned that in the last of that paper Thordarson/Hibbert do note that the F-test alone may not be sufficient; the scatter plot should always be inspected and perhaps more importantly for the current manuscript they state that “If certain binding constants in multi-species

equilibria are very low, e.g. K_2 in the 1 : 2 host-guest equilibria, the information content associated with that complex is inherently very limited. No method, no matter how sophisticated is likely to yield any reliable estimate of that binding constant (they will have large uncertainties). " I will come back to this point in terms of the results in the current paper but the authors might also mentioned in Figure 1, 'inspection of the scatter plots' and 'question models with very low K values".

1b) If one follows more recent work from Thordarson, e.g. JACS 2019, 141, 20146, he also suggested reporting "as is" the results for more than 1 binding models in cases where there is no obvious data to reject one of the author. Thordarson also uses in that paper the "Bayesian Information Criteria" (BIC – see page S30 of that paper) as tool to compare binding models rather than the F-test but BIC is related to AIK.

1c) Meanwhile Douglas Van Griend has published a number of papers that apply Factor analysis and bootstrap methods for model selections, including J. Chemometrics, 2022, 36, e3409 and references cited - this paper also gives a good overall historical perspective on this topic.

2) It would help the reader that is evaluating the results to see all the $K/K_1/K_2/SS$ values for the individual "triplicate" experiments displayed in Figure 3 and S3. As discussed in Thordarson/Hibbert 2016 the repeatability of the experiment as analyzed by standard dev / overall uncertainty (error) is useful in model comparison. Hence this raw data would be very useful (even in an excel file).

3) Linked to point no 2 – what are the fitted chemical shifts for the 1:1 viz 1:2 results? Are any of them "physically meaningless"? If so, then that might be used as further evidence that one of the models is probably not appropriate here.

4) In Figure 4 – have the free energy values (ΔG) or $\ln(K)$ been corrected for statistical factors, i.e. the K values in the 1:2 model results correct by: dividing K_1 with 2 and multiplying K_2 with 2. If the K values for the 1:2 model have not been corrected for statistical factors (see e.g. JACS, 2014, 136, 7505). This could then impact on the van't Hoff analysis in Figure 4.

5) The biggest concern I have with this work is how small the K values are! Figure 3 shows that for the 1:1 model the association constants (K_a) are about $3 \times 10^{-2} = 0.03 \text{ M}^{-1}$ for both hosts. In terms of dissociation constant (K_d) that is $K_d = 1/K_a = 33.3 \text{ M}$! For 1:1 model K_d is roughly the concentration required to reach a 50% of the host. 33.3 M of CHCl_3 is about 3 g per mL of chloroform-benzene solutions or about 2.8 mL of CHCl_3 per mL of chloroform-benzene solution which is clearly not physically possible. So either there is a very significant error in these calculations or how the results are presented here, because as presented they make little sense.

6) Somewhat related to that is then also K_1 and K_2 values. They not only vary widely as the authors do note, but they are also very small. Which setting aside the issues raised in point 5 here (but they also apply) should be a very strong warning signal as pointed out by Thordarson/Hibbert in 2016 (see point 1a above). So, even if there is some sort of a systematic error here that explains the strange numbers discussed in point 5 above, the fact that K_1 is often very small and in some cases that K_1 is much smaller than K_2 (e.g. Fig S3 in some of the examples) should already have been a big concern.

7) Although I find it unlikely here, have the authors role out that hosts 1a / 1b can dimerize / aggregate? If the hosts did it would complicate the data analysis further.

8) Setting all the above aside, one weakness of the van't Hoff approach is that it does assume the enthalpy (ΔH) and entropy (ΔS) is not changing (much) with temperature. This is of course only true for "ideal" systems and likely to fail say if the temperature range is large or say electrostatic / charge interactions play a significant role. In this particular case we have non-polar solvents and fairly non-polar host/guest so the system might be more likely to be in the "ideal" regime although the very high guest concentrations required (see also point 5) might counteract here. In any case, the authors

should at least note this potential issue with the van't Hoff analysis.

9) In conclusion, I think the van't Hoff analysis could be a good additional tool to rule out questionable binding model given that a "robust" binding model should (taking though my point 8 into consideration) follow van't Hoff behavior. My main concerns relate to i) some data issues (no 5 in particular but also 2, 3, 4, 6 and 7) and ii) that the authors may need to make a better note of the strengths of some other recent approach (no 1) and the potential weakness of their own approach (no 7). The messaging though should be clear – model selection is not trivial and ideally, the system being investigated should be studied with more than one technique, at more than one different concentration of the host, at more than one temperature or some combination of these methods.

Reviewer #3 (Remarks to the Author):

From fundamental studies to a huge array of downstream applications, the discovery and study of supramolecular complexation events remains a high interest area in the chemical and materials sciences. The author has previously published a number of influential papers advocating for further rigor when it comes to characterizing such complexes, particularly with respect to the stoichiometry of the binding partners. This study further builds on that backdrop, showing v'ant Hoff analysis to provide an important validation step. Overall, this is a detailed and high-quality study, which provides a novel contribution to the field. Given the potential for wide impact of the data within the broad supramolecular community, I think Nat Commun. is an excellent and appropriate forum. I am strongly supportive of acceptance. I did not spot any significant issues with the data and, as such, have relatively few comments. However, I would ask for the authors to consider the following points:

1) In the discussion, the authors propose a procedure for future stoichiometry analyses: 1) Fit with multiple models; 2) check GOF; 3) check using P-value or w_i values; 4) do v'ant Hoff. In this particular study the outcome of 3) and 4) were in disagreement. This provides an issue for the usage of this protocol. Perhaps the authors could further hone their guidance? What should be done when different approaches give different answers?

2) Related to point 2, it would be nice to have further insight on why there is a disconnect between the best fitted model and v'ant Hoff for the complex under study.

3) This manuscript studies a single supramolecular complex. Could the authors further comments on the validity of their protocol for other complexes (particularly tighter binders)?

Point-by-point responses. Our responses are shown in blue.

REVIEWER COMMENTS

Reviewer #1 (Remarks to the Author):

In this manuscript, Isobe et al propose a new, rigorous workflow to examine the validity of equilibrium models. They put forward van't Hoff analyses for effective validation of stoichiometry when investigating the hydrocarbon cage hosts and their chloroform complexes. The authors have synthesized an intriguing molecule and described its encapsulation mechanism with chloroform beautifully. Apart from experimental observations, theoretical studies were carried out to further confirm the stoichiometry of these complexes. Overall, this work is intriguing, all experiments were well conducted and the conclusion drawn is an important one – we indeed need to be critical of our findings with only one type of measurement for stoichiometry experiments. For works of methodology like the one here, there is naturally a question of novelty. However, it is my strong belief that proposing universal methods that are simple to grasp and execute has tremendous value to the scientific method. I believe the work to be thorough, the method well described, and recommend publication in principle.

We wish to thank this reviewer for finding our work intriguing and also for his/her evaluation with kind suggestions to improve our work.

Nevertheless, I see a few flaws that I would like to get the author's comments on:

1. Universality. Their current manuscript demonstrates the method on «only two» host-guest complexes.

Once again, we thank this reviewer for expecting our novel method to be universal.

Let us first summarize our present work:

1. Design and synthesis of a new cage molecule (**1a**) via a 5-step synthetic route (shortest count) from a commercially available compound. Another reference cage molecule (**1b**) was also synthesized via an identical route.
2. Establishing a molecular structure of **1a** by crystallographic analysis.
3. Titration experiments with **1a**. 13 spectra were obtained for one titration set, which was performed at 6 different temperatures with 3 runs for each titration to record $13 \times 6 \times 3 = 234$ ^1H NMR spectra in total.
4. Titration experiments with **1b**. 13 spectra were obtained for one titration set, which was performed at 6 different temperatures with 3 runs for each titration to record $13 \times 6 \times 3 = 234$ ^1H NMR spectra in total.
Thus, in total for these "only two" complexes, 468 ^1H NMR spectra were recorded.
5. Fitting analyses of 468 spectra by using GOF, F -test/ P -value and AIC/w_i value.
6. van't Hoff validation of K values from 6 different temperature conditions, respectively, with two compounds, **1a** and **1b**.

7. Theoretical DFT calculations and quantum mechanical AIM analyses to observe reasonable matching with the crystal structure.

As this reviewer recognizes, our present work on a novel cage compound along with a reference support compound has demonstrated that this method can be potentially universal. As a start, we believe that this work succeeds in marking an important conceptual advance. We also wish to ask this reviewer to consider our extensive experimentations and theoretical works behind this conceptual advance, derived from these "only" yet "important" two complexes. Experimentations and theoretical investigations of the present work are not narrow but rather extensive for one communication paper, as summarized above.

A conceptual advance is an important trigger that ignites follow-up works to confirm (or disprove) the universality, and the conceptual advance does not necessarily accompany actual demonstrations of universality in full. The demonstration of universality normally requires a lot of examples (not just by original discoverers but also by others) and needs to go through the test of time.

For instance, Makoto Fujita's first nanometer-sized supramolecular cage

(<https://doi.org/10.1038/378469a0>), albeit with a few examples, ignited follow-up works of metal-cornered supramolecular cages, and O. M. Yaghi's first, single example MOF

(<https://doi.org/10.1038/46248>) is still stimulating many of modern examples of MOF series. When the works are judged as important by others, conceptual advances can naturally stimulate many follow-up studies for the universality to be proven with many examples. Although we cannot be 100% sure about the future impact of the present work, we do believe that our present work can at least stimulate important discussion in the field of supramolecular chemistry. We hope that many others can be interested in designing (or searching for) novel host-guest systems to investigate the modern methodologies of stoichiometry inference.

In addition, the result from this manuscript has provided a counter-example to the previous paper published by the same group early this year (*Angew. Chem. Int. Ed.* 2023, 62, e2022190).

Yes. This is correct, and this fact shows the current status of the stoichiometry inference: this subject is premature and is still calling for further studies.

We did hope that our previous method with AIC could be decisive, but the method was found inapplicable to the present case. However, we feel lucky to find another method, derived from a very different approach at a different stage. As discussed below, having many different measures (reasons) is important to infer the credible stoichiometry of host-guest complexes.

I do think the manuscript would benefit from a few additional examples, even if it is reanalyses of previously published manuscripts – from the authors or others.

We wish to thank this reviewer for his/her kind suggestion. First of all, let us describe an unfortunate fact for the studies of host-guest equilibrium: there are little raw data available for re-evaluations by others. At present, unfortunately, we do not find any appropriate raw data for reanalyses with our methodology. Secondly, the van't Hoff data are getting less and less available due to its time-consuming processes (temperature-dependent analyses) and a rise of an alternative, powerful/concise method, *i.e.*, isothermal titration calorimetry (ITC). As this reviewer may kindly recognize, we have indeed published many papers on host-guest equilibria. However, most of them were analyzed by ITC and not by van't Hoff analyses. Therefore, we cannot readily demonstrate the universality. Nonetheless, after publication of this manuscript as the first case, we are determined to start investigating other cases for scope-and-limitation studies.

2. Methodology. I am not entirely convinced that I need to follow the workflow proposed in Fig.1. Why do we have to go for model comparisons (F-test/P-value; AIC/wi)?

You do not have to go through the whole processes of the workflow unless you have a reason to do so. Our attitude may be something like this:

"Based on our chemical intuitions/experiences, we come up with an idea for an equilibrium stoichiometry. Do we have any evidence to support this model (at any stage of analyses)?" Until you find a convincing support, you can follow the proposed workflow.

There are many stages as the checkpoint for the examination of your hypothesis. We summarize our workflow in detail below as a table. In an extreme case, you could even stop at the stage 2 and use the R^2 values for comparisons of stoichiometry models, only if these models are not "nested" (*e.g.*, see <https://doi.org/10.1002/anie.202219059>). If the models are nested, you must go forward to the stage 3. For the comparison at this stage 3, you can pick any measures with different grounds (F -test and/or AIC). Up to this stage was proposed in our previous study, but here in the present case, no available measure supported a chemically reasonable model (*i.e.*, 1:1). We therefore invented a novel method to verify a stoichiometry model at a different, new stage (*i.e.*, stage 5), with classic thermodynamics as a ground. It was indeed necessary and successful to use van't Hoff analyses as a measure in this case. We hope that this invention could be used by others to help them conclude their credible stoichiometry model.

stage	scientific ground	method
0. spectrum fitting	spectroscopy	soft (present)/hard (modern)*
1. isotherm fitting	thermodynamics	$\Delta\delta = f(K_n)$
2. GOF of isotherm fitting	statistics	R^2
3. comparison of isotherm fitting	statistics	F -test/ P -value
3. comparison of isotherm fitting	information-theoretic	AIC/ w_i value
4. K_n values	thermodynamics	from stage 1
5. van't Hoff validation	thermodynamics	$1/T$ -ln K plot

*For details, see responses to Reviewer #2.

Let us describe our own examples:

In our previous paper appeared in *Angewandte Chemie* 2023 (<https://doi.org/10.1002/anie.202219059>), AIC/ w_i was proven useful to support a chemically reasonable model. In our paper appeared in *Chem. Asian J.* 2022, F -test/ P -value (and other qualitative measures) supported reasonable models (<https://dx.doi.org/10.1002/asia.202200076>). Here in this study, there was no support for a chemically reasonable model until we introduced a novel, van't Hoff validations. Before these recent examples, the Job plot was used. Most of the examples had additional supports such as crystal structures and/or theoretical models.

Considering discussion on this point, we revised the Discussion part to give a clearer guide for the readers. We wish to thank this reviewer for his/her comments to improve this point.

In the cases presented in this manuscript, model comparisons did not help and gave the wrong conclusion. I think it should be made abundantly clear whether F/P -tests of AIC/ w_i should always be conducted or abandoned in favor of van't Hoff validation. In case of contradicting conclusions, it should also be made clear, under which circumstances which approach should be given more consideration. A guideline for the workflow is described as above, and we believe that the detailed description can be helpful for the reviewers and readers. NB: If we do not have any support for one (hypothesized) stoichiometry model, we cannot adopt it as a credible model due to the lack of evidence/support. Researchers can examine various measures such as F -test/ P -value, AIC/ w_i and van't Hoff validations as much as they want to examine the credibility of the hypothesized model. In addition, the readers should note that the credibility analysis with F -test/ P -value has also a statistical problem, and we added one paper for this discussion in the reference: Anderson, D. R., Burnham K. P. & Thompson, W. L. Null hypothesis testing: Problems, prevalence, and an alternative. *J. Wildl. Manag.* **64**, 912-923 (2000).

The most important fact is as follows: there is no single decisive method for the stoichiometry inference, and we need to examine the credibility of the stoichiometry model with various measures when necessary. At least in the present work, without introduction of new van't Hoff validations, a chemically reasonable 1:1 model was not supported by any of known measures. We hope that this novel method can also help others to find a support for their hypothesized, reasonable stoichiometry models.

Reviewer #2 (Remarks to the Author):

This paper introduces a new approach to deal with one of the most difficult challenges in the field of host-guest binding studies – namely “model selection” or selecting which binding model describes best the data being analyzed in supramolecular titration studies. This topic is of on-going interest and debate in the field and given that the accurate determination of binding constants and real stoichiometry in supramolecular underpins any attempts for the rational develop of better host-guest systems, this topic is clearly of interest to the readership of Nature Communication. The authors key insight and contribution in this paper resides in using van't Hoff analysis for validation of binding stoichiometries. I do think that this approach could be a very valuable tool in the field but there are a few issues in the current manuscript that first needed be addressed before the paper is published.

We wish to thank this reviewer for finding our work interesting and valuable for readers and also for his evaluation with kind suggestions to improve our work.

1. To start with, the historical overview in the introduction is very useful, including Figure 1.

We wish to thank this reviewer for finding our introduction "*very useful*".

But there are few points that may need adjustment, particular in relation to the 2016-2023 period:

1a) The authors frequently cite the Thordarson/Hibbert paper from 2016 and the F-test discussed in that review. But they don't mentioned that in the last of that paper Thordarson/Hibbert do note that the F-test alone may not be sufficient; the scatter plot should always be inspected and perhaps more importantly for the current manuscript they state that “If certain binding constants in multi-species equilibria are very low, e.g. K_2 in the 1 : 2 host–guest equilibria, the information content associated with that complex is inherently very limited. No method, no matter how sophisticated is likely to yield any reliable estimate of that binding constant (they will have large uncertainties). “ I will come back to this point in terms of the results in the current paper but the authors might also mentioned in Figure 1, ‘inspection of the scatter plots’ and ‘question models with very low K values”.

We wish to thank this reviewer for his careful reading. We may have simplified too much of the background. We thus revise the sentence of concern as follows:

A statistical method for the *F*-test using the *P*-value measure is useful to afford quantitative evaluations for the model comparison analyses, which has been suggested by Hibbert/Thordarson among various

measures. More recently, we introduced an information-theoretic method using Akaike's information criterion (AIC) with the Akaike weight (w_i) measure as a more versatile method for comparison. We also modified a relevant text in the main text.

1b) If one follows more recent work from Thordarson, e.g. JACS 2019, 141, 20146, he also suggested reporting "as is" the results for more than 1 binding models in cases where there is no obvious data to reject one of the author. Thordarson also uses in that paper the "Bayesian Information Criteria" (BIC – see page S30 of that paper) as tool to compare binding models rather than the F-test but BIC is related to AIK.

We wish to thank this reviewer for bringing this paper (Sessler/Thordarson/Gong; *AAAA–DDDD Quadruple H-Bond-Assisted Ionic Interactions: Robust Bis(guanidinium)/Dicarboxylate Heteroduplexes in Water*) to our attention.

Let us first summarize the relevant facts:

It is interesting to note that Thordarson very briefly mentioned BIC in his review (Chem. Commun. 2016; see above), "*However model selection is a mature statistical field and information theory gives approaches (e.g. Bayesian Information Criterion) that could be applied here. We describe below simple F-value calculations to aid the choice of model.*", but did not demonstrate its applicability until this JACS paper of 2019. Then, this JACS 2019 paper is rather complicated. No conclusion was indeed made on the stoichiometry (it could be 1:1, 1:2, 2:1 or mixed equilibrium): "*The data obtained was fitted to several binding models, including 1:1, 1:2 and 2:1 host–guest binding interactions, as well as a more complex 2:1 ↔ 1:1 ↔ 1:2 host–guest complexation model.*" They reported all the data for 1:1, 1:2 and 2:1. This is a very different approach from ours. We believe that making conclusion for stoichiometry with decent amount of discussion of quantitative measures and chemical reasons should be necessary. Nonetheless, this discussion is about the "stage 3" of the workflow (see below and/or our responses to Reviewer #1), which is not the central issue of the present work. As was discussed in our previous paper for AIC (<https://doi.org/10.1002/anie.202219059>), modern information-theoretic approaches including multimodel inference should be more appropriate (e.g., Burnham, K. P. & Anderson, D. R. *Model Selection and Multimodel Inference: A Practical Information-Theoretic Approach*, Springer, New York, 2nd edn (2002)). Interested readers can examine a sophisticated approach of multimodel inference described in the Chapter 4 of this book. Although we do not know how to apply this multimodel inference to subjects in the field of supramolecular chemistry yet, we believe that the multimodel inference could be one of the effective solutions to the stoichiometry inference in complicated cases, which was also supported by this example of Sessler/Thordarson/Gong.

We then summarize important points related to our present study:

Importantly, this JACS 2019 paper was not explicitly about the stoichiometry inference, as its title shows. In the main text of this paper, quantitative measures and discussion about the model selection do not appear and, for instance, the important term of BIC has never appeared in the main text. This is because the stoichiometry inference is not the central subject of this paper, unlike the present work of ours. Therefore, we feel it inappropriate to discuss further on this work in our present manuscript. Nonetheless, for the reader of the present manuscript to glance at a relevant method (despite the absence of decent discussion on the credibility), we wish to make this material, *i.e.*, the reviewer comments and our responses, openly accessible.

1c) Meanwhile Douglas Van Griend has published a number of papers that apply Factor analysis and bootstrap methods for model selections, including J. Chemometrics, 2022, 36, e3409 and references cited - this paper also gives a good overall historical perspective on this topic.

We thank this reviewer for introducing a method that was unfamiliar to us. However, this comment originates from misconceptions of this reviewer. We carefully read this paper and found that this work is not relevant to the present work.

A summary table of stages for titration experiments is shown below. Van Griend's paper is about the stage 0.

stage	scientific ground	method
0. spectrum fitting	spectroscopy	soft (present)/hard (modern)*
1. isotherm fitting	thermodynamics	$\Delta\delta = f(K_n)$
2. GOF of isotherm fitting	statistics	R^2
3. comparison of isotherm fitting	statistics	F -test/ P -value
3. comparison of isotherm fitting	information-theoretic	AIC/wi value
4. K_n values	thermodynamics	from stage 1
5. van't Hoff validation	thermodynamics	$1/T$ -ln K plot

The paper in question was titled "A reliable algorithm for calculating stoichiometry parameters in the hard modeling of spectrophotometric titration data" and reported a new method for so-called "hard modeling". The "hard modeling" is a method to resolve each component of a photometric spectrum, such as a UV-vis spectrum, containing multiple species by fitting analyses (same term but different meaning; e.g. see Kriesten, E., Alsmeyer, F. Bardow, A. & Marquardt, W. Fully automated indirect hard modeling of mixture spectra. *Chemomet. Intell. Lab. Sys.* **91**, 181-193 (2008).). First, unfortunately, this method can never be applied to fast-exchanging-regime, time-averaged NMR spectra that are the

spectral data in the present work. After resolving each component, Van Griend evaluated the fitting credibility with a primitive measure such as root-mean-square errors, which is in a distance from this work that is carefully examining the credibility of models with various statistic/informatic/thermodynamic methods. Most importantly, it is a tool for a different stage of titration experiments (stage 0 vs stage 5). Therefore, we do not find this work, including introduction, related to the present manuscript.

2) It would help the reader that is evaluating the results to see all the K/K1/K2/SS values for the individual “triplicate” experiments displayed in Figure 3 and S3. As discussed in Thordarson/Hibbert 2016 the repeatability of the experiment as analyzed by standard dev / overall uncertainty (error) is useful in model comparison. Hence this raw data would be very useful (even in an excel file).

We wish to thank this reviewer for bringing our attention to this point. To confirm the repeatability, we now performed independent fitting analyses for the three runs of titrations and showed the averaged association constants with standard deviations (SD) in the Methods section. The SD values were considerably low particularly for 1:1 models.

We also provide all raw, source data [$\Delta\delta = f([\text{CHCl}_3]_0/[1]_0)$] in Supplementary Information (Source data) as excel files, as suggested.

3) Linked to point no 2 – what are the fitted chemical shifts for the 1:1 viz 1:2 results? Are any of them “physically meaningless”? If so, then that might be used as further evidence that one of the models is probably not appropriate here.

We wish to thank this reviewer for this comment. Indeed, the fitted chemical shifts from the 1:1 models were consistent for both **1a** and **1b**, whereas the fitted chemical shifts from the 1:2 models considerably differed between **1a** and **1b**. Thus, as this reviewer kindly suggested, the fitted chemical shifts provided an additional indication for the 1:1 models. This fact and related data were included in the revision.

4) In Figure 4 – have the free energy values (DG) or $\ln(K)$ been corrected for statistical factors, i.e. the K values in the 1:2 model results correct by: dividing K1 with 2 and multiplying K2 with 2. If the $\ln K$ values for the 1:2 model have not been corrected for statistical factors (see e.g. JACS, 2014, 136, 7505). This could then impact on the van't Hoff analysis in Figure 4.

We wish to thank this reviewer for bringing the statistical factors to our attention. By considering the nature of the present host as a one-site host, we introduced necessary statistical factors to the analyses and revised the data accordingly. Along with our own example for the one-site host [Matsuno, T., Takahashi, K., Ikemoto, K. & Isobe, H. Activation of positive cooperativity by size mismatch assembly via inclination of guests in a single-site receptor. *Chem. Asian J.* **17**, e202200076 (2022)], we added another reference [Cavatorta, E., Jonkheijm, P. & Huskens, J. Assessment of cooperativity in ternary peptide-cucurbit[8]uril complexes. *Chem. Eur. J.* **23**, 4046-4050 (2017)].

5) The biggest concern I have with this work is how small the K values are! Figure 3 shows that for the 1:1 model the association constants (K_a) are about $3 \times 10^{-2} = 0.03 \text{ M}^{-1}$ for both hosts. In terms of dissociation constant (K_d) that is $K_d = 1/K_a = 33.3 \text{ M}$! For 1:1 model K_d is roughly the concentration required to reach a 50% of the host. 33.3 M of CHCl_3 is about 3 g per mL of chloroform-benzene solutions or about 2.8 mL of CHCl_3 per mL of chloroform-benzene solution which is clearly not physically possible. So either there is a very significant error in these calculations or how the results are presented here, because as presented they make little sense.

Albeit small, the association constants were derived by using well-established standard procedures, and the quality of the data were warranted by measures such as GOF for the fitting and R^2 values for the van't Hoff analyses. In addition, the repeatability was further warranted by performing titration in triplicate. Conversion of association constants to dissociation constants might have helped the reviewer postulate actual experimentations (such as 3 g/mL CHCl_3 in benzene [*sic*]), but we do not actually need to increase the concentration up to such an impossible regime. Such rhetorical considerations cannot deny the quantitative evidence provided by the statistics (GOF) and thermodynamics (R^2 for van't Hoff). We provided necessary data as grounds of discussion, and discussion needs to be made on these grounds with quantitative/scientific meanings and values. A claim like "*The values are small, and I cannot believe them.*" cannot make review processes constructive and productive.

Nonetheless, as the editors have discussed in their Editorial, we believe that *Nature Communications* should provide an extremely ideal place for the present work by opening not only the comments that "*represent valuable scholarship*" but also a record of disagreements such as this case: "*we also feel that in some cases – where authors and referees could not reach complete agreement on how certain data should be interpreted – making reports and the response from the authors public shows that valid potential caveats or limitations of the study have been raised and discussed. Peer review, in itself, cannot completely guarantee the validity of the conclusions reached by authors, especially in situations where competing hypotheses exist. Armed with the peer review file, each reader can better critically assess the robustness of the conclusions. Making the back and forth between authors and referees available also benefits authors, as it allows them to bring forward arguments in support of their view that might otherwise be difficult to logically integrate within a paper's narrative. –Editorial at Nature Communications (<https://doi.org/10.1038/s41467-022-33056-8>).*"

6) Somewhat related to that is then also K_1 and K_2 values. They not only vary widely as the authors do note, but they are also very small. Which setting aside the issues raised in point 5 here (but they also apply) should be a very strong warning signal as pointed out by Thordarson/Hibbert in 2016 (see point 1a above). So, even if there is some sort of a systematic error here that explains the strange numbers

discussed in point 5 above, the fact that K1 is often very small and in some cases that K1 is much smaller than K2 (e.g. Fig S3 in some of the examples) should already have been a big concern.

This comment is repeating the same claim as for the point #5 and saying that this reviewer cannot believe the presented values without providing scientific reasons. Let us repeat our message: our experimental values were unequivocally supported by the quantitative evidence provided by statistics (GOF) and thermodynamics (R^2 for van't Hoff) with a reasonable support for the repeatability of experiments (triplicate titrations). As reasonable scientific grounds have been provided by the authors for their results/conclusion, we believe that scientific grounds/evidence/references must be provided by the reviewer for his claims to deny experimental facts/results/conclusion.

7) Although I find it unlikely here, have the authors rule out that hosts 1a / 1b can dimerize / aggregate? If the hosts did it would complicate the data analysis further.

This is an unreasonable, groundless comment. This reviewer by himself knows the lack of grounds for his unreasonable claims, as can be seen from his leading statement of "*I find it unlikely here*". For such comments without any scientific grounds/evidence/references, it is impossible to exchange any further scientific, constructive discussion.

As this reviewer surely recognizes, making "possible" hypotheses is the most important first step of stoichiometry inference. Dimerization of the host cannot possibly be an ideal hypothesis. If he wishes to further discuss on this point, possible structures imagined by the reviewer should be at least provided, hopefully, with a decent reason to dream about such an imaginary structure.

8) Setting all the above aside, one weakness of the van't Hoff approach is that it does assume the enthalpy (ΔH) and entropy (ΔS) is not changing (much) with temperature. This is of course only true for "ideal" systems and likely to fail say if the temperature range is large or say electrostatic / charge interactions play a significant role. In this particular case we have non-polar solvents and fairly non-polar host/guest so the system might be more likely to be in the "ideal" regime although the very high guest concentrations required (see also point 5) might counteract here. In any case, the authors should at least note this potential issue with the van't Hoff analysis.

This is a serious misconception of the reviewer. The temperature-independent ΔH and ΔS are not at all an issue/weakness but rather a fundamental framework of the van't Hoff theory. This is one of the most important grounds of the van't Hoff theory that can be found in any textbook dealing with the theory, and we do not think it necessary to call attention to this point. The present work was developed within the framework of van't Hoff's (the 1st Nobel prize winning) laws of chemical dynamics. As we carefully described its background in the Introduction section with references, interested readers can readily go back to original literatures. As an example for inexperienced readers, we may quote relevant sentences from "MacQueen, J. T. Some observations concerning the van't Hoff equation. *J. Chem. Educ.* **44**, 755-

756 (1967)": "If ΔC_p° , the heat capacity of all products in their standard states minus the heat capacity of all reactants in their standard states, is zero for a given range of T , then ΔH° and ΔS° will be independent of T over this range and eqn. (5) indicates that a plot of $\ln K_a$ versus $1/T$ should be a straight line from whose slope ΔH° may be evaluated in the usual way." We are all "standing on the shoulders of giants" and always need to consider the grounds/frameworks for scientific discussion. If this manuscript can be accepted, this material, *i.e.*, the reviewer comments and our responses, will be made available to readers, and the readers can at least know the presence of temperature-dependent approaches for the enthalpy/entropy (which is not directly related to the present work).

9) In conclusion, I think the van't Hoff analysis could be a good additional tool to rule out questionable binding model given that a "robust" binding model should (taking though my point 8 into consideration) follow van't Hoff behavior. My main concerns relate to i) some data issues (no 5 in particular but also 2, 3, 4, 6 and 7) and ii) that the authors may need to make a better note of the strengths of some other recent approach (no 1) and the potential weakness of their own approach (no 7). The messaging though should be clear – model selection is not trivial and ideally, the system being investigated should be studied with more than one technique, at more than one different concentration of the host, at more than one temperature or some combination of these methods.

This is not an independent point of concern but rather a summary of his/her comments. We are happy to learn that this reviewer agrees in that "*the van't Hoff analysis could be a good additional tool*". All the points of concerns were carefully considered and/or rebutted as described above.

Reviewer #3 (Remarks to the Author):

From fundamental studies to a huge array of downstream applications, the discovery and study of supramolecular complexation events remains a high interest area in the chemical and materials sciences. The author has previously published a number of influential papers advocating for further rigor when it comes to characterizing such complexes, particularly with respect to the stoichiometry of the binding partners. This study further builds on that backdrop, showing v'ant Hoff analysis to provide an important validation step. Overall, this is a detailed and high-quality study, which provides a novel contribution to the field. Given the potential for wide impact of the data within the broad supramolecular community, I think Nat Commun. is an excellent and appropriate forum. I am strongly supportive of acceptance. I did not spot any significant issues with the data and, as such, have relatively few comments. However, I would ask for the authors to consider the following points:

We wish to thank this reviewer for being supportive of acceptance of our manuscript and also for his/her evaluation with kind suggestions to improve our work.

1) In the discussion, the authors propose a procedure for future stoichiometry analyses: 1) Fit with multiple models; 2) check GOF; 3) check using P-value or w_i values; 4) do van't Hoff. In this particular study the outcome of 3) and 4) were in disagreement. This provides an issue for the usage of this protocol. Perhaps the authors could further hone their guidance? What should be done when different approaches give different answers?

This is a question that is similar to the point #2 of Reviewer #1, and we avoid repeating the same message here and try to rephrase it to deepen our discussion.

Unfortunately, there is no single method that can be decisive, but if one stoichiometry model finds no support from the available measures, we should not choose such a model for the equilibrium in question. When different approaches give different answers, we may need to look for other supports/reasons based on chemistry/structures/theory. It was the case of our present example: all the previous measures such as F -test/ P -value and AIC/w_i did not support the 1:1 model which possessed the most reasonable structure. If we did not have the new, van't Hoff validation, we would not be able to select the most probable, 1:1 model for the equilibrium.

Considering discussion on this point, we revised the Discussion part to "hone our guidance". We wish to thank this reviewer for his/her comments to improve this point. We also hope that the further detailed procedures described in response to the point #2 of Reviewer #1 could help the reviewers and readers understand the workflows.

2) Related to point 2, it would be nice to have further insight on why there is a disconnect between the best fitted model and van't Hoff for the complex under study.

All the measures are not scientifically related each other. In this sense, they are all independent and disconnected. Each measure looks at the equilibrium from their scientific grounds, and the stages of evaluations can also be different. A brief summary of this fact is shown below:

stage	scientific ground	method
0. spectrum fitting	spectroscopy	soft (present)/hard (modern)*
1. isotherm fitting	thermodynamics	$\Delta\delta = f(K_n)$
2. GOF of isotherm fitting	statistics	R^2
3. comparison of isotherm fitting	statistics	F -test/ P -value
3. comparison of isotherm fitting	information-theoretic	AIC/w_i value
4. K_n values	thermodynamics	from stage 1
5. van't Hoff validation	thermodynamics	$1/T$ -ln K plot

*For details, see responses to Reviewer #2.

Thus, there is no rationale that can explain how differences in stoichiometry models appear.

3) This manuscript studies a single supramolecular complex. Could the authors further comments on the validity of their protocol for other complexes (particularly tighter binders)?

We thank this reviewer for this question. We hope that many more examples can be accumulated for the van't Hoff validations in the future, which should clarify the needs as well as scope and limitation of the present method. We will surely examine more examples in the future and hope that the readers who are reading this material could also join such studies.

List of revisions (highlighted with yellow backgrounds in the text).

[main text]

page 2: Correction of background descriptions. In response to Reviewer #2.

page 5: Correction of background descriptions. In response to Reviewer #2.

page 5: Correction of AIC-related values, originated from reanalyses with statistical factors. In response to Reviewer #2.

page 6: Correction of Fig. 3, originated from reanalyses with statistical factors. In response to Reviewer #2.

page 7: Correction of statistical factors. In response to Reviewer #2.

page 8: Correction of Fig. 4, originated from reanalyses with statistical factors. In response to Reviewer #2.

page 10: Modifications of guides in the Discussion section. In response to Reviewers #1 and #3.

page 12: Addition of experimental facts. In response to Reviewer #1.

page 12: Comments on newly provided source data. In response to Reviewer #2.

page 12-13: Addition of fitting data in Supplementary Information. In response to Reviewer #2.

page 13: Correction of statistical factors. Additional data of SD. In response to Reviewer #2.

page 13: Additional description about the repeatability. In response to Reviewer #2.

page 13: A comments on the fitted chemical shifts. In response to Reviewer #2.

page 13: Source data availability in Data in availability section. In response to Reviewer #2.

page 14, Ref. 7: Correction of title.

page 14, Ref. 14: An additional reference. In response to Reviewer #1.

page 15, Ref. 35 and 36: Additional references. In response to Reviewer #2.

[Supplementary Information]

page 6: correction of description to increase readability

page 8: additional descriptions on SS values. In response to Reviewer #2.

page 8: additional descriptions on fitted chemical shifts. In response to Reviewer #2.

page 9-10: additional data in Supplementary Figs. 3 and 4. In response to Reviewer #2.

page 10: Correction of eq. 1 (typos).

page 12: Correction of Supplementary Fig. 5, originated from reanalyses with statistical factors. In response to Reviewer #2.

Supplementary Figures were renumbered.

raw source data of titration experiments with **1a**. In response to Reviewer #2.

raw source data of titration experiments with **1b**. In response to Reviewer #2.

REVIEWER COMMENTS

Reviewer #1 (Remarks to the Author):

I stand by my initial assessment: this is an important paper and should be published. The current revision is clearer and I thank the authors for thoroughly discussing each raised point. I do like the table (top of page 4 of the rebuttal) and I would suggest including it in the paper.

Regarding the scope: I fully understand that an assessment as presented here takes a lot of time and more examples are not urgently needed for this manuscript - especially when (re)interpretation of "old" data is not readily possible. It would be good, however, to include a statement in the manuscript to underline why these systems are "hallmark" examples and provide solid ground from which to extrapolate the use of the methodology to other systems.

Considering these very minor comments, I do agree for the manuscript to be accepted.

Reviewer #2 (Remarks to the Author):

I like to thank the authors for their responses to the comments that I made, as well as those from other reviewers.

I still think that the idea of using van't Hoff as an additional tool for the model selection problem in supramolecular titration studies is a very important advance for the field and this work therefore has a lot going for it.

However, if anything, I am now more concerned than I was previously about certain aspects of the current manuscript. Perhaps it all boils down to that this particular host-guest system is not well suited as a proof of concept for the van't Hoff approach. I will give a more point by point response to the author's rebuttals, however, the three key issues that need to be addressed are - and I will then expand on further below:

A. The binding of CHCl_3 to the hosts 1a and 1b is very weak indeed. The data can be fitted to 1:1 and 1:2 binding model but the results are meaningless as the chemical shifts for the H in CHCl_3 in the bound complexes are physically impossible, ranging from +64 ppm to -100 to -300 ppm.

B. There is also potentially some issues with the binding equilibria equations used and/or how Origin is doing the data regression.

Details on A: The authors have given more details on their results, although that could have been done in a more user-friendly way, both when it comes to the host/guest concentrations and more importantly, the obtained chemical shifts. Starting with the latter, Figures S3 and S4 appears to show the limiting $\delta(\text{bound})$ values. For the 1:1 complexes (Fig S3) then it appears that $\delta(\text{bound}) = -275$ ppm and -260 ppm, for 1a and 1b, respectively. For the 1:2 complexes (Fig S4) the $\delta(\text{bound HG})$ and $\delta(\text{bound HG}_2) = -112$ ppm and $+64$ ppm viz -345 ppm and 0.1 ppm, respectively, for 1a and 1b.

A very ruffled porphyrin paramagnetic compound has been reported with a proton resonance around -58 ppm (In Org. 2004, 43, 5034) and based on a Wikipedia article (https://en.wikipedia.org/wiki/Transition_metal_hydride) the $\text{IrHCl}_2(\text{PMe}(\text{t-Bu})_2)_2$ complex has a 1H at -50.5 ppm. Chemical shifts of $+100$ or -100 ppm in the current case seem extremely unlikely.

This is directly related to my previous concerns about only a very small portion of the CHCl_3 guest

being bound at any given time if the K-values are indeed as low as reported here ($< 0.1 \text{ M}^{-1}$). Fig S3 and Fig S4 also seem to recognize that less than 1% of the CHCl_3 is bound.

Hence the argument that the GOF is good is not very useful – if the model gives physically impossible chemical shifts then clearly something is wrong with it.

I tried to recreate the binding data based on the information given. Unfortunately, the excel files supplied do not give the host / guest concentration but only their ratio = x . When I tried to use the information in the experimental to recreate the host / guest concentration and x , I could not get exactly the same values as the others. I used the assumption that the initial volume is 690 μL and that when you add $1+3+4+4+6+7+25+50+100+100+100+200+200 \mu\text{L}$ (see page 12 of the paper) then the final volume should be 1490 μL . With initial amount of $1a = 3.47 \mu\text{mol}$ then the concentration of $1a = 2.33 \text{ mM}$ when the final addition of the CHCl_3 has been done. And if the concentration of CHCl_3 in the “guest solution” is 1.01 M and a total of $1490 - 690 = 800 \mu\text{L}$ of CHCl_3 have been added times $1.01 \text{ M} = 8088 \mu\text{mol} / 1490 \mu\text{L} = 542 \text{ mM}$. This would give $x = 542 / 2.33 = 232$ and not 322 as per excel spreadsheet. More details in the excel spreadsheets would have eliminated this issue.

Assuming, however, that the x values are correct in the excel spreadsheet given, I used the volumes above and the $1a$ concentration to recalculate the CHCl_3 at each data point and then use this as an input on the supramolecular.org (bindfit) website using the $1a$ data at 283 K. I also “flipped” the host/guest definition and put the CHCl_3 concentration = host (this seems correct given the titration is following the changes in the chemical shift of the CHCl_3 - see also point B below). The results are interesting:

1:1 complex - see:

<http://app.supramolecular.org/bindfit/view/902ddb5d-c318-4d6a-9fa2-8bed83f1662f>

$ss = 1.5404\text{e-}6$, $K = 0.32$, $\Delta(\text{bound}) = -29 \text{ ppm}$

1:2 complex - see:

<http://app.supramolecular.org/bindfit/view/946caa65-b4f1-4144-8855-e73d43f89c36>

$ss = 1.5339\text{e-}6$, $K_1 = 0.35$ and $K_2 = 37.29$, $\Delta(\text{bound HG}) = -31 \text{ ppm}$ and $\Delta(\text{bound HG}_2) = +20 \text{ ppm}$

Noting that the input here isn't probably an exact copy of what the authors used here, this exercise is still noteworthy in that bindfit returns:

- i) Very low K-values (< 0.1) suggesting saturation of the “guest” never exceeds 1%.
- ii) Unrealist $\Delta(\text{bound})$ values.

In addition, the bindfit results give almost the same ss values (c.a. $1.5\text{e-}6$) for the 1:1 and 1:2 model and in both cases lower than in Figure S5a. More details on the input / output numbers from the authors would be necessary to ascertain to why but maybe it is related to some rounding issue differences between the Origin program and the python engine used by bindfit. Such rounding issues would normally not matter but given how weak the interaction is here (if any?) it is perhaps not as surprising.

Another problem with both the author's and bindfit's 1:2 results is that $K_2 \gg K_1 =$ very strong positive cooperativity. This means that the data is very information poor when it comes to the $\Delta(\text{bound HG})$ values or indeed anything re what is happening in the formation of HG if a 1:2 binding process is taking place.

Summary re point A: Irrespective of whether one assumes 1:1 or 1:2 binding, the interaction here is simply too weak to be meaningfully measured. The saturation (fraction of HG bound) never exceeds

1%. Even more worryingly, the $\Delta(\text{bound})$ values obtained are physically impossible.

Details on B: I mentioned above that I “flipped” the host/guest definitions for CHCl_3 and 1a in my attempt to fit the data with bindfit . As per reference 6 in the current manuscript, the definitions of host and guest are somewhat arbitrary. As also discussed in that reference, it is the mole-fraction that matters with NMR data and normally one would define the guest as “silent” – and hence, following reference 6, I “flipped” the definitions of host and guest in my attempt to use bindfit . But in the context of the author’s work then this brings up another big concern: How did the authors adjust their fitting process to the fact that they are fitting the data to what they call the “Guest” = CHCl_3 resonance? Because Eq. 1 on page 12 shows $[\text{H}]_0$ in the denominator? Is that then 1a even if the authors are then monitoring the CHCl_3 resonance?

And to add to the confusion here Eq 1 (And Eq 2) do not look anything like the Origin program code on page S6. It would be helpful if the exact connection between Eq 1 and Eq 2 and the Origin code was clearer as well as if the authors can check whether they should be using the mole-fraction of the bound host (1a) to calculate changes in the observed resonance for bound/unbound CHCl_3 ? My final suggestion below could perhaps help with resolving some of these issues.

I do also have some concerns and questions about some of the other answers in the rebuttal:

Q1c: I am not sure it is right of the authors to dismiss so easily the Vander Griend paper. I encourage the authors to re-read the introduction to their paper (section 1.2), which shows that related approaches have been explored by many other authors, in particular with the work of Maeder mentioned. This approach is yes, fundamentally different to the F-statistics, AIK, BIC etc... but has the great advantage of being nearly “unbiased”. Yes, Vander Griend only uses it for UV-Vis but why couldn't also be adopted for NMR analysis?

Q2: As mentioned above, although it was helpful to get the excel files they do not include the absolute concentrations for CHCl_3 and 1a/1b. While I am at it, excel files with the raw numbers for the AIK and van't Hoff calculations / plots would not hurt. And the chemical shifts ($\Delta(\text{bound})$ etc) need to be tabulated in the SI rather than just shown in the Figures there.

Q3: See main point A above. This is the biggest problem with the manuscript.

Q4: I am going to assume from the answer that the statistical factors were indeed included for the 1:2 analysis in the van't Hoff plot but the raw calculations in an excel file would help here too.

Q5: As per main point A in particular (and to some extent point B) – fitting the data to either the 1:1 or 1:2 gives physically meaningless ^1H NMR shifts. Which on top of my previous concerns suggests this host-guest interaction is simply too weak to measure.

Q6: Saying the data fits well to a model (and hence gives a good GOF) does not mean the results have any real physical meaning. Which as per my main point A above is the main issue here (not whether the GOF is good or not).

Q7: I made this comment thinking the authors were following the chemical shift from the host (1a or 1b) and hence it wouldn't be hard to see if those chemical shifts would change with the concentration of these hosts. Now that it is clearer that the authors are following the ^1H resonance for CHCl_3 in C_6D_6 then it yes, seems a bit less likely that CHCl_3 could be dimerizing in C_6D_6 . No need though for the authors not to check this by measuring the CHCl_3 shift at different concentrations in C_6D_6 without the “host” 1a/1b present. See also my final suggestion below.

It is however also noteworthy that during the course of the titration the mole ratio of C6D6/CHCl3 changes from ca 7700 to 20. If there is a significant CH- π interaction between CHCl3 and C6D6 could that also generate the ca 0.05 ppm shift observed during the titration?

Q8: In the authors response to my original point number 8, it seems like the authors cannot conceive the possibility that the van't Hoff relation is not linear. This is despite the fact that even Wikipedia (! - https://en.wikipedia.org/wiki/Van_'t_Hoff_equation) recognizes that "However, in some cases the enthalpy and entropy do change dramatically with temperature." Importantly, in the context of the current manuscript such deviations have been noted both ITC (calorimetry) host-guest interactions:

Biophysical Chemistry, 2004, 110, 15, calorimetric vs van't Hoff binding enthalpies from isothermal calorimetry: Ba²⁺-crown ether complexation. E.g. Figure 1 which shows that ΔH is not constant with temperature and Figure 5 which shows non-linearity of the van't Hoff plot. Thus it is not unreasonable to raise the question about the expected linearity in the van't Hoff plot here – although such a discussion is probably not very fruitful anyway, given the issues with the binding constants obtained (See points A and B).

J. Am. Chem. Soc. 2008, 130, 9, 2798 also states: "A body of literature has emerged that suggests that the enthalpic and entropic parameters for molecular recognition are highly correlated for a wide variety of phenomena.[44-49] Such high correlations, with R values often exceeding 0.95, would seem to suggest some common chemical phenomena behind the forces governing molecular recognition for a wide range of systems. However, in spite of the prevalence of these enthalpy–entropy compensation effects in the literature, they remain controversial and many authors have noted that their appearance may be artifactual.[50-56] This is because indirect methods, such as van't Hoff analysis, do not independently measure the ΔH and ΔS of an equilibrium. For measurements spanning only a small temperature range, any true determination of the ΔH and ΔS may be obscured by a dominant statistical correlation that arises from the linear least-squares regression.[53,54] Additionally, the assumption in van't Hoff analysis that ΔH is constant (i.e., the heat capacity change is negligible) may be invalid.

Therefore, I think it not unreasonable to have asked the question if ΔH constant here or not. Although considering my main points A and B now this is a secondary issue. And yes, just because van't Hoff plots sometimes fail because ΔH isn't perhaps temperature independent, doesn't mean that the van't Hoff approach couldn't be useful for model selection. I just think it needs to be noted that van't Hoff can fail for the reasons mentioned in the above papers.

Q9: As I said now in Main point A and B – I have now even more serious concerns that the data obtained from fitting the titration data to the 1:1 or 1:2 binding model has any real physical meaning. Other factors, such as CH- π (in CHCl3 to C6D6) cannot be ruled out. And in any case, even if 1a/1b are indeed binding to CHCl3 the interactions are simply too weak to measure accurately with ¹H NMR it seems. To prove the point that van't Hoff plot help with model selection, a stronger binding system might be desirable.

As a final suggestion to the authors – why not titrate a solution of 1a (or 1b) into a solution of CHCl3 in C6D6. Ideally by making up the 1a (or 1b) solution in the "host" CHCl3/C6D6 solution so that there is no dilution of CHCl3 during the course of the titration? Further, in doing that titration collect both a significant number of titration point in the range of 0 to 2 eq and also beyond 10 eq of 1a (or 1b). This would probably make it easier to measure the very weak interaction between 1a (or 1b) and CHCl3 and also explore if a 1:2 (1 molecule CHCl3 + 2 molecules 1a or 1b) or a 2:1 (2 molecules CHCl3 + 1 molecule 1a or 1b) binding models do indeed fit better to the data than the 1:1. This experiment would only have to do be done at one temperature (e.g. 298 K) and the models compared with the AIK method.

Reviewer #3 (Remarks to the Author):

The authors have responded robustly to all the reviewer comments. Further detail has been added to the paper to address several of the concerns. I still believe this is an important piece of work and certainly worthy of publication in Nat Commun. However, have two further contributions to the discussion:

1) Reviewer 2 raised several critical comments on the work. In my opinion, the author's response to these comments was more aggressive than it needed to be. Nonetheless, I am specifically not satisfied with their response to comment 5. Reviewer 2 asked for the author to explain the physical meaning of a very small equilibrium constant. The author's response, as far as I understand it, was to say that their data is robust. This is dodging the question. Earlier in their responses, the author highlights the value of chemical intuition, whether to generate a hypothesis on a given binding stoichiometry, or to sanity check the models. I think this is all that reviewer 2 was after; a sanity check on a very low value. My recommendation would be that the authors state, explicitly, that they either cannot give a meaningful physical interpretation, or else provide their interpretation. This would not require any additional data.

2) All reviewers asked for examples of the van't Hoff analysis being applied to supramolecular complexes beyond the two highly related complexes studied in this paper. I agree with the author that they should not be expected to generate additional data on a comprehensive set of additional complexes. However, what I think the reviewers want is reassurance that the analysis shown in this paper is relevant beyond the very specific example system provided. I don't think this is an unreasonable request and I still suggest that at least one additional example is provided. This could be reanalysis of a known complex (reducing the workload). From their prior work, the author presumably has many reagents available that could be used to provide an additional example. This would just leave the NMR time, which is significant, but not unreasonable.

Point-by-point responses. Our responses are shown in green. In the main text and Supplementary Information, our revisions are highlighted with colored backgrounds (yellow = 1st round, blue = 2nd round).

We wish to thank all the Reviewers for their kind efforts to examine our work again and for their valuable comments. Taking account of their constructive suggestions, we revised our manuscript to a considerable extent by performing additional experiments and re-analyses. Two major revisions were made:

1. A second case from re-analyses of reported titration data was found to confirm the applicability of the van't Hoff validation method to other systems. (In response to the Reviewers #1 and #3)
2. A new congener with different substituents was synthesized to confirm the consistent, large $\delta_{\Delta\text{HG}}$ values. (In response to the Reviewer #2)

We hope that the Reviewers can find the revised manuscript sufficiently improved and find our work interesting and suitable for publication.

Major additional data are listed below.

[List of additional data]

Additional data 1: Synthesis of a new molecule **1c**.

Additional data 2: Titration experiments with **1c** and chloroform from variable-temperature ^1H NMR spectra (234 spectra).

Additional data 3: Fitting analyses with **1c** \supset CHCl_3 for GOF (R^2), AIC (w_i) and van't Hoff validations. The fitting analyses also confirmed the consistent $\Delta\delta_{\text{HG}}$ values from 1:1 models with **1a**, **1b** and **1c**.

Additional data 4: Re-analysis of reported titration data (252 ^1H NMR spectra) with phenine [5]circulene (**5**) and C_{60} .

Additional data 5: Fitting analyses with **5** \supset C_{60} for GOF (R^2), AIC (w_i) and van't Hoff validations. The fitting analyses confirmed the applicability of the van't Hoff validation method to other systems.

Additional data 6: A list of $\delta_{\Delta\text{HG},n}$ values in Supplementary Information (Supplementary Tables 2-4) with related discussion.

In total, 954 ^1H NMR spectra were analyzed in this study.

* In addition to corrections in response to reviewers, minor corrections (e.g. typos) found by us were also made.

REVIEWER COMMENTS

Reviewer #1 (Remarks to the Author):

I stand by my initial assessment: this is an important paper and should be published. The current revision is clearer and I thank the authors for thoroughly discussing each raised point. I do like the table (top of page 4 of the rebuttal) and I would suggest including it in the paper.

We would like to thank this reviewer to find our work important. The table is included as Supplementary Table 5 in Supplementary Information after minor modification to remove the irrelevant stage 0.

Regarding the scope: I fully understand that an assessment as presented here takes a lot of time and more examples are not urgently needed for this manuscript - especially when (re)interpretation of "old" data is not readily possible. It would be good, however, to include a statement in the manuscript to underline why these systems are "hallmark" examples and provide solid ground from which to extrapolate the use of the methodology to other systems.

We wish to thank this reviewer for his/her comments on the scope. We finally succeeded in obtaining our variable-temperature titration data from our previous study (<http://dx.doi.org/10.1021/acs.orglett.7b00899>) and performed re-analyses to examine the scope of the van't Hoff validations. We excavated 252 ^1H NMR spectra that were recorded at our previous affiliation, Tohoku University, during September 2016 and reanalyzed them for the fitting analyses. As detailed in our revised manuscript, GOF (R^2) and AIC (w_i) did not necessarily support the 1:1 host-guest complex that was observed in the crystal structure, and the van't Hoff validations supported this chemically reasonable structure. We believe that the results now demonstrate a wide applicability of the present van't Hoff validation method.

Considering these very minor comments, I do agree for the manuscript to be accepted.

Once again, we wish to thank this reviewer for his/her kind evaluation.

Reviewer #2 (Remarks to the Author):

I like to thank the authors for their responses to the comments that I made, as well as those from other reviewers.

I still think that the idea of using van't Hoff as an additional tool for the model selection problem in supramolecular titration studies is a very important advance for the field and this work therefore has a lot going for it.

We wish to thank this reviewer for finding our work important.

However, if anything, I am now more concerned than I was previously about certain aspects of the current manuscript. Perhaps it all boils down to that this particular host-guest system is not well suited as a proof

of concept for the van't Hoff approach. I will give a more point by point response to the author's rebuttals, however, the three key issues that need to be addressed are – and I will then expand on further below:
A. The binding of CHCl₃ to the hosts 1a and 1b is very weak indeed. The data can be fitted to 1:1 and 1:2 binding model but the results are meaningless as the chemical shifts for the H in CHCl₃ in the bound complexes are physically impossible, ranging from +64 ppm to -100 to -300 ppm.

We describe our responses below at the place where his detailed comments appear.

B. There is also potentially some issues with the binding equilibria equations used and/or how Origin is doing the data regression.

We describe our responses below at the place where his detailed comments appear.

Details on A: The authors have given more details on their results, although that could have been done in a more user-friendly way, both when it comes to the host/guest concentrations and more importantly, the obtained chemical shifts.

We found that the amount of the mother aliquot, "0.500 mL", was missing by mistake in the Method section and apologize for this. We provided raw data for the line graphs, and the file format was following guidelines given by the Editors in their email. As recommended by the Editor, we uploaded titration data at figshare. In the revision, we also provided further details including instruments so that the readers can also examine possible sources of errors.

Starting with the latter, Figures S3 and S4 appears to show the limiting delta(bound) values. For the 1:1 complexes (Fig S3) then it appears that delta(bound) = -275 pm and -260 pm, for 1a and 1b, respectively. For the 1:2 complexes (Fig S4) the delta(bound HG) and delta(bound HG₂) = -112 pm and +64 ppm viz -345 pm and 0.1 ppm, respectively, for 1a and 1b.

A very ruffled porphyrin paramagnetic compound has been reported with a proton resonance around -58 ppm (In Org. 2004, 43, 5034) and based on a Wikipedia article (https://en.wikipedia.org/wiki/Transition_metal_hydride) the IrHCl₂(PMe(t-Bu)₂)₂ complex has a 1H at -50.5 ppm. Chemical shifts of +100 or -100 ppm in the current case seem extremely unlikely.

This is directly related to my previous concerns about only a very small portion of the CHCl₃ guest being bound at any given time if the K-values are indeed as low as reported here (< 0.1 M⁻¹). Fig S3 and Fig S4 also seem to recognize that less than 1% of the CHCl₃ is bound.

Hence the argument that the GOF is good is not very useful – if the model gives physically impossible chemical shifts then clearly something is wrong with it.

I tried to recreate the binding data based on the information given. Unfortunately, the excel files supplied do not give the host / guest concentration but only their ratio = x. When I tried to use the information in the experimental to recreate the host / guest concentration and x, I could not get exactly the same values

as the others. I used the assumption that the initial volume is 690 uL and that when you add 1+3+4+4+6+7+25+50+100+100+100+200+200 uL (see page 12 of the paper) then the final volume should be 1490 uL. With initial amount of 1a = 3.47 umol then the concentration of 1a = 2.33 mM when the final addition of the CHCl₃ has been done. And if the concentration of CHCl₃ in the “guest solution” is 1.01 M and a total of 1490 – 690 = 800 uL of CHCl₃ have been added times 1.01 M = 8088 umol / 1490 uL = 542 mM. This would give $x = 542 / 2.33 = 232$ and not 322 as per excel spreadsheet. More details in the excel spreadsheets would have eliminated this issue.

Assuming, however, that the x values are correct in the excel spreadsheet given, I used the volumes above and the 1a concentration to recalculate the CHCl₃ at each data point and then use this as an input on the [supramolecular.org\(bindfit\)](http://supramolecular.org/bindfit) website using the 1a data at 283 K. I also “flipped” the host/guest definition and put the CHCl₃ concentration = host (this seems correct given the titration is following the changes in the chemical shift of the CHCl₃ - see also point B below). The results are interesting:

1:1 complex

ss = 1.5404e-6, K = 0.32, delta(bound) = -29 ppm

1:2 complex

ss = 1.5339e-6, K₁ = 0.35 and K₂ = 37.29, delta(bound HG) = -31 ppm and delta(bound HG₂) = +20 ppm

Noting that the input here isn't probably an exact copy of what the authors used here, this exercise is still noteworthy in that bindfit returns:

- i) Very low K-values (< 0.1) suggesting saturation of the “guest” never exceeds 1%.
- ii) Unrealist delta(bound) values.

In addition, the bindfit results give almost the same ss values (c.a. 1.5e-6) for the 1:1 and 1:2 model and in both cases lower than in Figure S5a. More details on the input / output numbers from the authors would be necessary to ascertain to why but maybe it is related to some rounding issue differences between the Origin program and the python engine used by bindfit. Such rounding issues would normally not matter but given how weak the interaction is here (if any?) it is perhaps not as surprising.

Another problem with both the author's and bindfit's 1:2 results is that $K_2 \gg K_1$ = very strong positive cooperativity. This means that the data is very information poor when it comes to the delta(bound HG) values or indeed anything re what is happening in the formation of HG if a 1:2 binding process is taking place.

Summary re point A: Irrespective of whether one assumes 1:1 or 1:2 binding, the interaction here is simply too weak to be meaningfully measured. The saturation (fraction of HG bound) never exceeds 1%. Even more worryingly, the delta(bound) values obtained are physically impossible.

Before the titration experiments and van't Hoff validation, the 1:1 model of $1 \supset \text{CHCl}_3$ had 3 major supports: (1) The crystal structures of two congeners (**1a** and **1b**) showed that one chloroform molecule at the center was commonly observed at nearly identical positions. (2) DFT calculations showed that the

central chloroform molecule can have CH- π interactions with the cage host. (3) The crystal structure and the DFT structure matched almost perfectly.

These data thus formed our chemical intuition/hypothesis as: "the chloroform can form a 1:1 complex at the center of the cage host via CH- π hydrogen bonds". Importantly, the crystal/DFT structure show that the proton of the central chloroform molecule is located at the shielded region of the benzene panel, which should result in "upfield shifts" in ^1H NMR spectra with this 1:1 structure.

From the titration experiments, there were following important facts that were derived as experimental data and were consistent with those data and hypothesis described above: (4) Upfield shifts of chloroform were indeed observed by increasing the amount of chloroform against **1a** for 13 NMR spectra recorded at 298 K. (5) Almost identical upfield shifts of chloroform were observed with **1b** for 13 NMR spectra recorded at 298 K. (6) Experiments for points #4 and #5 were repeated 3 times to confirm the upfield shifts commonly observed to accumulate 78 upfield-shifted spectra for **1a** and **1b**. (7) The temperatures of spectra were varied for 6 data point for **1a** and **1b**, which persistently showed the upfield shifts of chloroform (accumulating 468 consistent spectra). (8) Because three measures of R^2/F -test/AIC failed to support the 1:1 model, the van't Hoff validation method was introduced and finally supported the 1:1 model. (9) An additional congener **1c** was synthesized, and the upfield shifts were again observed persistently over 117 additional spectra, *i.e.*, 704 upfield-shifted spectra were accumulated in total. (10) For **1c**, the van't Hoff validation also supported the consistent 1:1 model, which required those upfield-shifted spectra to be fitted with physically meaningful equations.

From these considerations, we derive the most important fact: The upfield shifts observed in 704 spectra suggested that chloroform proton was in the shielded region, which all agreed well with the crystal/DFT 1:1 structures. The crystal structures also indicate that a 2nd chloroform molecule (if any) in 1:2 structure can only be located at deshielded regions, which suggests downfield shifts to be observed.

The $\delta_{\Delta\text{HG}}$ values for 1:1 model for **1a**, **1b** and **1c** commonly showed upfield shifts with the similar values, which were consistent with the experimental spectra (upfield shifts). The $\delta_{\Delta\text{HG}}$ & $\delta_{\Delta\text{HG}2}$ values for 1:2 model for **1a**, **1b** and **1c** deviated to a large extent even ranging from upfield shifts to downfield shifts. These 1:2 data could not afford any physical picture for the interpretations.

The $\delta_{\Delta\text{HG}}$ values for 1:1 model were consistent with many other experimental facts as well as chemical intuitions. There remains only one mystery: Why can the $\delta_{\Delta\text{HG}}$ values be large in their degree? At present, we do not know, and it provides us an interesting subject to be investigated in the future.

The next important fact is that the small K_1 values ($3 \times 10^{-2} \text{ M}^{-1}$) were obtained from the titration experiments and were consistent with the DFT calculations suggesting small association energy ($\Delta E = -$

5.6 kcal/mol). Importantly, for such small K values to be observed, the δ_{AHG} values need to be large, as can be found in eq. S8, for instance: A minute amount of the complex can give rise to small yet observable physical changes of $\Delta\delta$, because of the large δ_{AHG} values.

We wish to thank this reviewer for those comments. Taking account of discussion here, we integrated these considerations and analyses with additional tables in the Supplementary Information (see below). We believe that these sections considerably reinforced our work. Please note that titration experiments with a newly synthesized congener **1c** and additional re-analyses of **5** and C_{60} systems (see below) were added for the discussion.

When we summarized the fitted values (δ_{AHG} for **1:1** and δ_{AHG} & δ_{AHG2} for **1:2**) in a table (Supplementary Table S2), the problems of 1:2 model fitting as well as the credibility of 1:1 fitting became clearer. The δ_{AHG} values for the 1:1 models were consistent for **1a-1c** throughout the experimental temperature range (283-328 K) and were obtained from -225 ppm to -296 ppm. The minor deviations were represented by small standard deviation (SD) values associated with the averaged δ_{AHG} values. On the other hand, the δ_{AHG} and δ_{AHG2} values for the 1:2 models considerably varied. For example, the δ_{AHG} values for **1a** ranged from -8001 ppm to -110 ppm. The δ_{AHG2} values varied from downfield to upfield shifts even for one molecule: a large deviation from -562 ppm to +4.5 ppm was suggested for **1b**. Although we do not understand the origin of the large $|\delta_{\text{AHG}}|$ values for the 1:1 model, these values were experimentally meaningful within a framework of the present titration experiments. Additional data of relative standard deviations in comparison with re-analyses of $\mathbf{5}_n \supset (\text{C}_{60})_m$ further reinforced the conclusion (see below). Importantly, the large $|\delta_{\text{AHG}}|$ values were essential to determine the small K_1 values. As eq. S8 suggests, for instance, it is the large $|\delta_{\text{AHG}}|$ value that makes a small physical change of $\Delta\delta$ from a minute amount of the complex detectable/observable during the titration. The anomalous large values provide an interesting subject to be investigated in the future.

* * *

When we compared δ_{AHGm} values of "**1** + CHCl_3 " and "**5** + C_{60} " in detail, we found additional data that supported the credibility of the δ_{AHG} values for $\mathbf{1} \supset (\text{CHCl}_3)_1$. From the average values with standard deviations, we can derive the relative standard deviations (%RSD) that are commonly used as a standardized measure of dispersion. The %RSD values for four systems were calculated for the average values and were shown in Supplementary Table 4. The fitting with a reasonable model (1:1) showed a small dispersion in a range of 2-8%, but the fitting with unreasonable models (1:2/2:1) showed a large dispersion such as 468% (1:2 of **1a** + CHCl_3) at the maximum. Thus, these values confirmed the credibility of δ_{AHG} values for the "**1** + CHCl_3 " systems.

As for the reviewer comments on the 1:2 model, we also had a relevant sentence in the main text about its theoretical study and found that the fact might not be clearly conveyed. We corrected a relevant sentence as follows: "As shown in Fig. 5, the structure of **1:1** was obtained from a geometry optimization exhibiting good convergence, whereas the structure of **1:2** with two chloroform molecules did not converge." We do not find any chemical reason that can support a larger association constant for the second chloroform guest.

Details on B: I mentioned above that I “flipped” the host/guest definitions for CHCl₃ and 1a in my attempt to fit the data with bindfit. As per reference 6 in the current manuscript, the definitions of host and guest are somewhat arbitrary. As also discussed in that reference, it is the mole-fraction that matters with NMR data and normally one would define the guest as “silent” – and hence, following reference 6, I “flipped” the definitions of host and guest in my attempt to use bindfit. But in the context of the author’s work then this brings up another big concern: How did the authors adjust their fitting process to the fact that they are fitting the data to what they call the “Guest” = CHCl₃ resonance? Because Eq. 1 on page 12 shows [H]0 in the denominator? Is that then 1a even if the authors are then monitoring the CHCl₃ resonance? And to add to the confusion here Eq 1 (And Eq 2) do not look anything like the Origin program code on page S6. It would be helpful if the exact connection between Eq 1 and Eq 2 and the Origin code was clearer as well as if the authors can check whether they should be using the mole-fraction of the bound host (1a) to calculate changes in the observed resonance for bound/unbound CHCl₃? My final suggestion below could perhaps help with resolving some of these issues.

We thank this reviewer for careful examinations. In this revision, we fully rewrote the fitting procedures to elaborate details and hope that the reviewers/readers can trace and reproduce our procedures. We also hope that the readers can utilize these non-black-box, transparent procedures and codes for their analyses through their own examinations of details.

I do also have some concerns and questions about some of the other answers in the rebuttal:

Q1c: I am not sure it is right of the authors to dismiss so easily the Vander Griend paper. I encourage the authors to re-read the introduction to their paper (section 1.2), which shows that related approaches have been explored by many other authors, in particular with the work of Maeder mentioned. This approach is yes, fundamentally different to the F-statistics, AIK, BIC etc... but has the great advantage of being nearly “unbiased”. Yes, Vander Griend only uses it for UV-Vis but why couldn't also be adopted for NMR analysis?

The paper in question is titled, "*A reliable algorithm for calculating stoichiometry parameters in the hard modeling of spectrophotometric titration data*", and as this title shows, this paper is about "spectrophotometric titration". As the basic equation for such photometric titrations is different from that for NMR, the paper in question cannot be directly applied to the present study. Below, we show a relevant paragraph from a tutorial review by Thordarson (doi: 10.1039/c0cs00062k):

The mathematical model used to obtain the association constant is usually developed from realising that the physical change (ΔY , e.g., a NMR shift or an change in UV-Vis absorbance) observed is correlated to the concentration of the complex [HG] as $\Delta Y \propto [HG]$, or in some cases, the free host [H] or the free guest [G]. The physical change (Y) being monitored can usually be described as the aggregate of the individual components according to eqn (7) as a function of concentration (e.g., for UV-Vis spectroscopy) or eqn (8) as a function of mole fractions f_X (f_X defined as: $f_X = [X]/[X]_0$) in the special case of NMR.

$$Y = Y_H[H] + Y_G[G] + Y_{HG}[HG] \quad (7)$$

$$Y = Y_{H/H} + Y_{G/G} + Y_{HG/HG} \quad (8)$$

Even if a short section (section 1.2; ca. 0.5 page) describes a literature survey in this paper in question, we do not find that the main subject of this 16-page paper is relevant to our present work. Therefore, we do not think it appropriate to cite this paper. Nonetheless, we found that similar analyses for NMR data were reported by Hübler and added a comment on this paper in Supplementary Information. This paper, Hübler, C. Analysing binding stoichiometries in NMR titration experiments using Monte Carlo simulation and resampling techniques. *PeerJ Anal. Chem.* **4**, e23 (2022), gives a nice overview and refers the Vander Griend's work. This should give correct perspectives over these methods to the readers.

Q2: As mentioned above, although it was helpful to get the excel files they do not include the absolute concentrations for CHCl_3 and 1a/1b. While I am at it, excel files with the raw numbers for the AIK and van't Hoff calculations / plots would not hurt. And the chemical shifts (delta (bound) etc) need to be tabulated in the SI rather than just shown in the Figures there.

We provided all the data and followed the guideline given by the editors for the format.

Q3: See main point A above. This is the biggest problem with the manuscript.

Please see our rebuttal above.

Q4: I am going to assume from the answer that the statistical factors were indeed included for the 1:2 analysis in the van't Hoff plot but the raw calculations in an excel file would help here too.

The equations were explicitly written in the main text. We also provided all the data and followed the guideline given by the editors for the format.

Q5: As per main point A in particular (and to some extent point B) – fitting the data to either the 1:1 or 1:2 gives physically meaningless ^1H NMR shifts. Which on top of my previous concerns suggests this host-guest interaction is simply too weak to measure.

Please see our rebuttal above.

Q6: Saying the data fits well to a model (and hence gives a good GOF) does not mean the results have any real physical meaning. Which as per my main point A above is the main issue here (not whether the GOF is good or not).

Please see our rebuttal above.

Q7: I made this comment thinking the authors were following the chemical shift from the host (1a or 1b) and hence it wouldn't be hard to see if those chemical shifts would change with the concentration of these hosts. Now that it is clearer that the authors are following the ^1H resonance for CHCl_3 in C_6D_6 then it yes, seems a bit less likely that CHCl_3 could be dimerizing in C_6D_6 . No need though for the authors not to check this by measuring the CHCl_3 shift at different concentrations in C_6D_6 without the "host" 1a/1b present. See also my final suggestion below.

Please see our comments below.

It is however also noteworthy that during the course of the titration the mole ratio of $\text{C}_6\text{D}_6/\text{CHCl}_3$ changes from ca 7700 to 20. If there is a significant CH- π interaction between CHCl_3 and C_6D_6 could that also generate the ca 0.05 ppm shift observed during the titration?

No, there could be no CH- π interactions between CHCl_3 and the solvent.

There is a very serious misconception of this reviewer. We never used benzene- d_6 (C_6D_6) but instead used cyclohexane- d_{12} (C_6D_{12}), which was consistently conveyed throughout the manuscript.

Nonetheless, to respect this reviewer comment, we now measured ^1H NMR spectra of CHCl_3 in cyclohexane- d_{12} (not C_6D_6) by varying concentrations and found no concentration-dependent shift.

Q8: In the authors response to my original point number 8, it seems like the authors cannot conceive the possibility that the van't Hoff relation is not linear. This is despite the fact that even Wikipedia (! - https://en.wikipedia.org/wiki/Van_t_Hoff_equation) recognizes that "However, in some cases the enthalpy and entropy do change dramatically with temperature." Importantly, in the context of the current manuscript such deviations have been noted both ITC (calorimetry) host-guest interactions:

Biophysical Chemistry, 2004, 110, 15, calorimetric vs van't Hoff binding enthalpies from isothermal calorimetry: Ba^{2+} -crown ether complexation. E.g. Figure 1 which shows that ΔH is not constant with temperature and Figure 5 which shows non-linearity of the van't Hoff plot. Thus it is not unreasonable to raise the question about the expected linearity in the van't Hoff plot here – although such a discussion is probably not very fruitful anyway, given the issues with the binding constants obtained (See points A and B).

J. Am. Chem. Soc. 2008, 130, 9, 2798 also states: "A body of literature has emerged that suggests that the enthalpic and entropic parameters for molecular recognition are highly correlated for a wide variety of phenomena.[44-49] Such high correlations, with R values often exceeding 0.95, would seem to suggest some common chemical phenomena behind the forces governing molecular recognition for a wide range of systems. However, in spite of the prevalence of these enthalpy–entropy compensation effects in the

literature, they remain controversial and many authors have noted that their appearance may be artifactual.[50-56] This is because indirect methods, such as van't Hoff analysis, do not independently measure the ΔH and ΔS of an equilibrium. For measurements spanning only a small temperature range, any true determination of the ΔH and ΔS may be obscured by a dominant statistical correlation that arises from the linear least-squares regression.[53,54] Additionally, the assumption in van't Hoff analysis that ΔH is constant (i.e., the heat capacity change is negligible) may be invalid. Therefore, I think it not unreasonable to have asked the question if ΔH constant here or not. Although considering my main points A and B now this is a secondary issue. And yes, just because van't Hoff plots sometimes fail because ΔH isn't perhaps temperature independent, doesn't mean that the van't Hoff approach couldn't be useful for model selection. I just think it needs to be noted that van't Hoff can fail for the reasons mentioned in the above papers.

We thank this reviewer for further clarifications. We added one sentence on the temperature-dependent enthalpy in the Discussion section with references. We believe that an additional reference to a textbook should be very useful for the readers: McQuarrie, D. A. & Simon, J. D. *Physical Chemistry: A Molecular Approach*. Chapter 26, 1049-1100 (University Science Books, Sausalito, 1997).

Q9: As I said now in Main point A and B – I have now even more serious concerns that the data obtained from fitting the titration data to the 1:1 or 1:2 binding model has any real physical meaning. Other factors, such as CH- π (in CHCl_3 to C_6D_6) cannot be ruled out.

For the responses and rebuttals to the two points, please see our comments above.

Nonetheless, there is a serious misconception of this reviewer. We never used benzene- d_6 (C_6D_6) and used cyclohexane- d_{12} (C_6D_{12}) throughout this study, which was consistently conveyed in the manuscript.

And in any case, even if 1a/1b are indeed binding to CHCl_3 the interactions are simply too weak to measure accurately with ^1H NMR it seems. To prove the point that van't Hoff plot help with model selection, a stronger binding system might be desirable.

To demonstrate the scope of the present van't Hoff validation method, we now included an additional case with phenine [5]circulene and C_{60} , as was suggested by the Reviewers #1 and #3. The results confirmed the applicability of the method to other systems.

As a final suggestion to the authors – why not titrate a solution of 1a (or 1b) into a solution of CHCl_3 in C_6D_6 . Ideally by making up the 1a (or 1b) solution in the “host” $\text{CHCl}_3/\text{C}_6\text{D}_6$ solution so that there is no dilution of CHCl_3 during the course of the titration? Further, in doing that titration collect both a significant number of titration point in the range of 0 to 2 eq and also beyond 10 eq of 1a (or 1b). This would probably make it easier to measure the very weak interaction between 1a (or 1b) and CHCl_3 and also explore if a 1:2 (1 molecule CHCl_3 + 2 molecules 1a or 1b) or a 2:1 (2 molecules CHCl_3 + 1 molecule 1a or 1b)

binding models do indeed fit better to the data than the 1:1. This experiment would only have to be done at one temperature (e.g. 298 K) and the models compared with the AIK method.

As this reviewer might also have conceived, such reversed titrations suffer from the solubility of the host: the concentration of **1** in cyclohexane needs to be increased up to around 1 M, which is not possible.

Nonetheless, we believe that we simply do not understand the origin of the large δ_{IHG} values that were consistently observed with three congeners (**1a**, **1b** and **1c**). The observed value now provides an interesting subject for us to investigate in the future. We wonder if dynamic single-axis rotations of the CHCl_3 guest could lead to unusual physical output (e.g. <https://doi.org/10.1038/s41467-018-04325-2>; <https://doi.org/10.1038/s41467-018-06270-6>; <https://doi.org/10.1038/s41467-021-25358-0>).

The additional valid case with phenine [5]circulene and C_{60} assessed the present van't Hoff validation method and confirmed its applicability to other cases.

Reviewer #3 (Remarks to the Author):

The authors have responded robustly to all the reviewer comments. Further detail has been added to the paper to address several of the concerns. I still believe this is an important piece of work and certainly worthy of publication in Nat Commun. However, have two further contributions to the discussion:

We wish to thank this reviewer for finding our work important and also for his/her valuable comments.

1) Reviewer 2 raised several critical comments on the work. In my opinion, the author's response to these comments was more aggressive than it needed to be. Nonetheless, I am specifically not satisfied with their response to comment 5. Reviewer 2 asked for the author to explain the physical meaning of a very small equilibrium constant. The author's response, as far as I understand it, was to say that their data is robust. This is dodging the question. Earlier in their responses, the author highlights the value of chemical intuition, whether to generate a hypothesis on a given binding stoichiometry, or to sanity check the models. I think this is all that reviewer 2 was after; a sanity check on a very low value. My recommendation would be that the authors state, explicitly, that they either cannot give a meaningful physical interpretation, or else provide their interpretation. This would not require any additional data.

We agree. We do not understand the origin of large δ_{IHG} values. In the previous revision, we described this fact in the footnote of the Supplementary Information, and in the present revision, we integrated the discussion and data in the main part of the Supplementary Information. Moreover, we synthesized another derivative with OMe substituents (**1c**) for the revision this time and performed the titration experiments likewise. The result showed that the very large δ_{IHG} values were consistently observed for the three congeners. We thus conclude that, although we do not understand the origin of these values,

they are meaningful values that can lead us to the observation of the small K values as well as the confirmation of van't Hoff validation methods.

2) All reviewers asked for examples of the van't Hoff analysis being applied to supramolecular complexes beyond the two highly related complexes studied in this paper. I agree with the author that they should not be expected to generate additional data on a comprehensive set of additional complexes. However, what I think the reviewers want is reassurance that the analysis shown in this paper is relevant beyond the very specific example system provided. I don't think this is an unreasonable request and I still suggest that at least one additional example is provided. This could be reanalysis of a known complex (reducing the workload). From their prior work, the author presumably has many reagents available that could be used to provide an additional example. This would just leave the NMR time, which is significant, but not unreasonable.

We thank this reviewer for this important comment. Finally, we found 7-year-old data from our previous study (<http://dx.doi.org/10.1021/acs.orglett.7b00899>). The experiments were performed in our previous affiliation, Tohoku University, and the data were not immediately available. Nonetheless, for the re-analyses, we excavated 252 VT ^1H NMR spectra for triplicate titrations. Importantly, as described in the main text, the re-analyses assessed the van't Hoff validation method for a different system and confirmed its applicability to this system with a higher K value. We wish to thank this reviewer for important and constructive suggestions and hope that he/she can find our work publishable.

REVIEWERS' COMMENTS

Reviewer #2 (Remarks to the Author):

I like to thank the authors once again for their responses.

This version of the manuscript is much improved! The data analysis for C60/5 system is much more convincing and better as an illustration of why using van't Hoff approach could be very useful in model selection.

In any case, with the raw data in hand it is now easier to fit the data, say with the bindfit program.

Doing that for the C60/5 system gives very similar results:

1:1 binding at 298 K (here the subtract initial value option is used as the first entry is host only)

Results (put the below url into a browser)

<http://app.supramolecular.org/bindfit/view/dbce0be0-c01b-4edf-ac4b-e2532afe3b81>

$K = 38977$ which matches the 39000 value in Figure S11 – 298 K

Interestingly though bindfit gives $ss = 3.07 \times 10^{-5}$ whereas Fig S11 shows $ss = 4.9 \times 10^{-5}$.

One can only assume that this due to Origin not finding the minima as accurately as the python engine in bindfit (but if one had the fitted Origin results also it would be easier to check).

This difference does not matter for the C60/5 but when it comes to the 1a, 1b and 1c with CHCl₃ the differences increase. For example, CHCl₃ + 1a at 298 K

1:1 binding (here CHCl₃ concentration is defined as "host" and an additional "dummy" entry for pure CHCl₃ and $\delta = 0$ ppm is added).

Results (put the below url into a browser)

<http://app.supramolecular.org/bindfit/view/9c6669fb-635a-4f87-b97d-61d724809d81>

$K = 0.09$ ($= 9 \times 10^{-2}$) which is a bit higher than 3×10^{-2} (0.03) in Figure S3 at 298 K

Here bindfit gives $ss = 5.38 \times 10^{-6}$ whereas Fig S11 shows $ss = 8.1 \times 10^{-6}$.

Here the $\delta(\text{bound})$ in bindfit is also ca -90 ppm is totally unrealistic although "slightly" better than the -282 ppm shown in Supp Table 2.

The point I am trying to make here is that because such a small fraction of the CHCl₃ is bound to 1a/1b/1c it is really not surprising that $\Delta(\text{bound})$ values for 1a/1b/1c are unrealistic as small differences in the minima found (as measured by ss) can also change the binding constant K by a factor 2-3 (from 0.03 to 0.09) whereas for the much tighter binding of C60 to 5 it does not matter much that Origin may not have found the exact same minima as the python program.

Therefore, I remain quite skeptical about the value of data analysis for 1a/1b/1c because the binding so weak.

But given the effort the authors have put in and given that the C60/5 system works a lot better to demonstrate their point, I would like to make the following final suggestion for adjustment to the manuscript (and I ok with the Editor checking if these are done without necessarily coming back to me if everything else is ok):

1. The fitted data, i.e. the one that generates the solid lines in say Fig 3 should also be included in the excel data files (see also previous comments).

2. Intermediate calculations of the AIC weights. I am still not 100% sure how these were calculated. A detailed example, e.g. in an excel spreadsheet would be useful.

3. Will the Origin fitting program file also be provided on Figshare (in line with modern Open Science practices)? Having the code in PDF is not as useful as having the actual origin file. Including the Origin raw input / output datafile would also be desirable.

4. At the moment the Main Article does NOT mention that the Delta(bound) values for 1a/1b/1c are very high and appear unrealistic. This need to be added both to the relevant part of the results and to the discussion. The reader needs to be aware of this issue with the 1a/1b/1c data without having to look into the SI. I don't mind if the rest of the paper stays the same.

All my other concerns and comments have been addressed.

Reviewer #3 (Remarks to the Author):

The authors have addressed all the comments robustly and added significant new data. I have no further comments and believe the paper should be published.

Point-by-point response. Our responses are shown in purple. In the main text, our revisions are highlighted with colored backgrounds (yellow = 1st round, blue = 2nd round, green = 3rd round).

REVIEWERS' COMMENTS

Reviewer #2 (Remarks to the Author):

I like to thank the authors once again for their responses.

This version of the manuscript is much improved! The data analysis for C60/5 system is much more convincing and better as an illustration of why using van't Hoff approach could be very useful in model selection.

We thank this reviewer for finding our manuscript improved.

In any case, with the raw data in hand it is now easier to fit the data, say with the bindfit program.

Doing that for the C60/5 system gives very similar results:

1:1 binding at 298 K (here the subtract initial value option is used as the first entry is host only)

Results (put the below url into a browser)

$K = 38977$ which matches the 39000 value in Figure S11 – 298 K

Interestingly though bindfit gives $ss = 3.07 \times 10^{-5}$ whereas Fig S11 shows $ss = 4.9 \times 10^{-5}$.

One can only assume that this due to Origin not finding the minima as accurately as the python engine in bindfit (but if one had the fitted Origin results also it would be easier to check).

This difference does not matter for the C60/5 but when it comes to the 1a, 1b and 1c with CHCl₃ the differences increase. For example, CHCl₃ + 1a at 298 K

1:1 binding (here CHCl₃ concentration is defined as “host” and an additional “dummy” entry for pure CHCl₃ and $\delta = 0$ ppm is added).

Results (put the below url into a browser)

$K = 0.09$ ($= 9 \times 10^{-2}$) which is a bit higher than 3×10^{-2} (0.03) in Figure S3 at 298 K

Here bindfit gives $ss = 5.38 \times 10^{-6}$ whereas Fig S11 shows $ss = 8.1 \times 10^{-6}$.

Here the $\delta(\text{bound})$ in bindfit is also ca -90 ppm is totally unrealistic although “slightly” better than the -282 ppm shown in Supp Table 2.

The point I am trying to make here is that because such a small fraction of the CHCl₃ is bound to 1a/1b/1c it is really not surprising that $\Delta(\text{bound})$ values for 1a/1b/1c are unrealistic as small differences in the minima found (as measured by ss) can also change the binding constant K by a factor 2-3 (from 0.03 to 0.09) whereas for the much tighter binding of C60 to 5 it does not matter much that Origin may not have found the exact same minima as the python program.

Therefore, I remain quite skeptical about the value of data analysis for 1a/1b/1c because the binding so weak.

But given the effort the authors have put in and given that the C60/5 system works a lot better to demonstrate their point, I would like to make the following final suggestion for adjustment to the

manuscript (and I ok with the Editor checking if these are done without necessarily coming back to me if everything else is ok):

We thank this reviewer for his detailed examinations with "bindfit". We believe that this site has been developed by Thordarson, and the web site states that its bases have been reported in his review (<https://doi.org/10.1039/C0CS00062K>). Some corrections of equations were also made in the literature (<https://www.rsc.org/suppdata/cs/c0/c0cs00062k/addition.htm>). However, even after these corrections, we still find incorrect equations (e.g. equations 11 and 22). The codes of the bindfit are provided in python codes, but we (synthetic chemists) cannot trace the errors and/or corrections in the long, complicated codes, unfortunately. Therefore, we are unavoidably afraid of introducing errors by using this site, considering the fact that errors exist in the original scientific paper. Under this situation, we could also presume/wonder if the difference between Origin and bindfit could originate from hidden errors in bindfit.

We believe that it is unavoidable for any data analyses to be deviated, depending on programs used for the analyses. For instance, crystallographic analyses can be performed by using programs such as Olex2, ShelXle or Yadokari-XG, and the results could have minor differences, even though the basic physics/equations are identical. The researchers surely need to clarify the programs and procedures, but they do not need to use one specific program. Even if one program is popular in a certain field, it should never be regarded as authoritative and standard to be used as a reference.

1. The fitted data, i.e. the one that generates the solid lines in say Fig 3 should also be included in the excel data files (see also previous comments).

We included the data. However, we hope that the journal can set a standard for data requirements in the future. We do not think that presenting such detailed raw data should be mandatory and necessary.

2. Intermediate calculations of the AIC weights. I am still not 100% sure how these were calculated. A detailed example, e.g. in an excel spreadsheet would be useful.

We believe that the detailed procedure including equation 20 in Supplementary Information should allow readers to follow our procedures. More importantly, AIC analyses have been reported in our preceding paper in details (<https://doi.org/10.1002/anie.202219059>) with excel files and detailed manuals. We do not think it necessary/ethical to repeat what has been previously reported in detail, and interested readers should also enjoy and examine the in-depth discussion about AIC in the original paper. Convenience/usefulness should not be set at the center of science.

3. Will the Origin fitting program file also be provided on Figshare (in line with modern Open Science practices)? Having the code in PDF is not as useful as having the actual origin file. Including the Origin raw input / output datafile would also be desirable.

We have provided codes in PDF and believe that this style is better than the proposed style to separate them. There are 5 independent (yet similar) codes for the present study, and for the readers to understand the content of the code, seeing them in context in the PDF file should be much easier. We do not understand "the Origin raw input/output" comments: The inputs were the raw data provided in excel files, and the outputs were results such as K and so on that were reported in the manuscript. They are all provided, and necessary items for reproduction can be found.

4. At the moment the Main Article does NOT mention that the Delta(bound) values for 1a/1b/1c are very high and appear unrealistic. This need to be added both to the relevant part of the results and to the discussion. The reader needs to be aware of this issue with the 1a/1b/1c data without having to look into the SI. I don't mind if the rest of the paper stays the same.

We added a following comment in the main text: As detailed in Supplementary Information, large $|\delta_{\text{HGL}}|$ values from fitting analyses indicated the presence of unknown origins in experimental values, which might be one of the reasons of the minor difference.

All my other concerns and comments have been addressed.

Reviewer #3 (Remarks to the Author):

The authors have addressed all the comments robustly and added significant new data. I have no further comments and believe the paper should be published.

We wish to thank this reviewer for supporting publication of our manuscript.